

# Spatial distribution of cloud droplet size properties from Airborne Hyper-Angular Rainbow Polarimeter (AirHARP) measurements

Brent A. McBride[1,2,3], J. Vanderlei Martins[1,2,3], Henrique M.J. Barbosa[4], William Birmingham[2,3], and Lorraine A. Remer[2,3]

[1]Department of Physics, University of Maryland Baltimore County, Maryland, USA
[2]Earth and Space Institute, University of Maryland Baltimore County, Maryland, USA
[3]Joint Center for Earth Systems Technology, University of Maryland Baltimore County, Maryland, USA
[4]Instituto de Física, Universidade de São Paulo, São Paulo, 05508-090, Brasil

**Correspondence**: Brent A. McBride (mcbride1@umbc.edu)

**Abstract.** The global variability of clouds and their interactions with aerosol and radiation make them one of our largest uncertainties related to global radiative forcing. The droplet size distribution (DSD) of clouds is an excellent proxy that connects cloud microphysical properties with radiative impacts on our climate. However, traditional radiometric instruments are information-limited in their DSD retrievals.
Radiometric sensors can infer droplet effective radius directly, but not the distribution width, which is an important parameter tied to the growth of a cloud field and to the onset of precipitation. DSD heterogeneity hidden inside large pixels, lack of angular information and the absence of polarization limits the amount of information these retrievals can provide. Next-generation instruments that can measure at
narrow resolutions, multiple view angles on the same pixel, with a broad swath, and sensitivity to intensity and polarization of light are best situated to retrieve DSDs at the pixel-level and over a wide spatial field. The Airborne Hyper-Angular Rainbow Polarimeter (HARP) is a wide field-of-view imaging polarimeter instrument designed by the University of Maryland, Baltimore County (UMBC) for retrievals of cloud droplet size distribution properties over a wide swath, narrow resolution, and at up to 60 unique, co-
located view zenith angles in the 670nm channel. Cloud droplet effective radius (CDR) and variance (CDV) of a unimodal gamma size distribution are inferred simultaneously by matching measurement to Mie polarized phase functions. For all targets with appropriate geometry, a retrieval is possible, and unprecedented spatial maps of CDR and CDV are made for cloud fields that stretch both across the swath and along the entirety of a flight observation. During the NASA Lake Michigan Ozone Study (LMOS)
aircraft campaign from May-June 2017, the Airborne HARP (AirHARP) instrument observed a heterogeneous stratocumulus cloud field along the solar principal plane. Our retrievals from this dataset show that cloud DSD heterogeneity can occur at the 200m scale, much smaller than the 1-2km resolution of most spaceborne sensors. This heterogeneity at the subpixel level can create artificial broadening of the DSD in retrievals made at resolutions on the order of 0.5 to 1 km. This AirHARP study demonstrates
the viability of the HARP concept to make cloud measurements at scales of individual clouds with global coverage, and all in a low-cost, compact CubeSat-size payload.



# 1 Introduction

Clouds are one of the most uncertain aspects of our climate system. Clouds are highly variable, yet well-dispersed across the globe and play a dual role in distributing energy: they trap infrared radiation in our atmosphere and reflect shortwave radiation back to space (Rossow and Zhang 1995). This energy distribution is the key unknown in predicting climate change, as the interplay between longwave trapping and shortwave reflection of radiation by clouds may change significantly as the planet warms. The relative strength of these impacts depend strongly on cloud macro- and microphysical properties, such as cloud optical thickness, thermodynamic phase, cloud top temperature, height, and pressure, liquid/ice water path and content, and droplet size distributions. Measuring these elements in a global context and over long temporal scales are crucial to improve our understanding of how cloud properties translate to a radiative impact on climate.

Clouds also share a delicate relationship with aerosols. Aerosols drop the energy barrier required for condensation, serving as *cloud condensation nuclei* (Petters and Kreidenweis 2007). These particles seed both liquid water and ice clouds in our atmosphere in a process that is so important to cloud development that without aerosols, clouds would not exist at the scale we see them today. When aerosols are entrained into a cloud, they can set off a condensation feedback loop but in some cases, the opposite occurs: they dry out the local atmosphere and evaporate smaller droplets (Hill, Feingold, and Jiang 2009, Small et al. 2009). Aerosols can invigorate convective clouds (Altaratz et al., 2014) and suppress the development of other clouds (Koren et al. 2004), depending on the aerosol and meteorological properties of the local atmosphere. This complexity is a major source of uncertainty related to understanding global radiative forcing and predicting climate change (Boucher et al., 2013, Rosenfeld et al. 2014, Penner et al. 2004, Coddington et al. 2010).

A major link between radiative and microphysical climate impacts of liquid water clouds is the droplet size distribution (DSD). A common mathematical representation of the liquid water cloud DSD is a gamma distribution (Tampieri and Tomasi 1975, Hansen 1971, Alexandrov et al. 2015). This DSD is formed of two parameters, cloud droplet *effective radius* (CDR or $r_{eff}$) and *effective variance* (CDV or $v_{eff}$, Hansen and Travis 1974), which represent the mean droplet size and dispersion relative to scattering cross-section. Aerosol effects on cloud microphysics are strongly tied to the CDR (Twomey 1977, Albrecht 1989). In a general example, aerosol loading generates competition for condensation sites and leads to smaller droplets. This process can delay rainout, but increase the overall liquid water content, extending the lifetime of the cloud. An abundance of smaller droplets scatter shortwave radiation efficiently, creating a brighter cloud, and finally the excess of radiation to space results in a net cooling of the planet (Haywood et al. 2000, Lohmann et al. 2000 and references therein). Typically, studies that connect microphysical and radiative properties of clouds do so by tracking changes in CDR only, with no direct sensitivity to CDV (Feingold et al., 2001, Platnick and Oreopoulos 2008). Because CDV is a measurement of the breadth of the DSD, it may encode information on cloud growth processes: collision/coalescence, aerosol or dry air entrainment, evaporation, and the initiation of precipitation on cloud cores or periphery. Not all clouds share the same relationship between microphysics and radiation, but the key to understanding the connection lies in the microphysics, described by these two DSD parameters. Only satellite instruments allow us to make these long-term connections between radiation and evolution of cloud DSDs for different cloud types and over large spatial and temporal periods. Also,



satellite studies best improve global models that examine both future climate scenarios and cloud feedbacks (Stubenrauch et al. 2013).

There are currently two methods used to retrieve CDR from spaceborne instruments. The first is the widely-used radiometric bi-spectral retrieval, first proposed by Nakajima and King (1990a) and employed operationally to MODerate resolution Imaging Spectroradiometer (MODIS) and other multi-band radiometer data (Platnick et al., 2003, 2017; Walther and Heidinger, 2012). The bi-spectral retrieval uses the difference in cloud information content observed by shortwave infrared (i.e. 1.6, 2.1, or 3.7 µm) and visible (i.e. 0.67 or 0.87 µm) channels to retrieve CDR and cloud optical thickness (COT) simultaneously for a cloud target. The second method is the multi-angle polarimetric retrieval, which is relatively new. The polarimetric retrieval corresponds to a parametric fit to a multi-angle polarized cloudbow (or rainbow by cloud droplets) structure that is sensitive to both CDR and CDV simultaneously (Breon and Goloub 1998, Alexandrov et al. 2015, Di Noia et al. 2019). COT can also be retrieved with assistance from an external radiative transfer simulation (Alexandrov et al. 2012a). 3D multiple scattering effects of shadowing and illumination (Marshak et al. 2006, Varnai and Marshak 2002) bias the radiometric method, whereas the polarimetric retrieval is sensitive to scattered photons from a COT up to ~3, lessening the impact of this effect (Miller et al. 2018). Subpixel clouds and spatial heterogeneities can affect both methods, as discussed in later sections (Zhang et al. 2011, Breon and Doutriaux-Boucher 2005, Shang et al. 2015). Furthermore, the bi-spectral technique is not sensitive to CDV and uses a pre-established value (0.1, Platnick et al. 2017) that may not be valid for all liquid water cloud targets and all regions of the world.

Multi-angle polarimetric measurements have other advantages for cloud characterization beyond retrieval of the two DSD parameters. We find that retrievals of cloud thermodynamic phase (Riedi et al. 2010, Goloub et al. 2000), ice crystal asymmetry (van Diedenhoven et al. 2012b), aerosol above cloud (Waquet et al. 2013), and COT (Xu et al. 2018, Cornet et al. 2018) are considerably improved with the addition of polarized observations. At the time of this writing, only the Polarization and Directionality of the Earth's Reflectances (POLDER, Deschamps et al. 1994) instruments have demonstrated polarized retrieval of cloud DSD properties from space, though several aircraft instruments, including the Airborne Multi-angle SpectroPolarimetric Imager (AirMSPI, Diner et al. 2013), the Research Scanning Polarimeter (RSP, Cairns et al. 1999), and the subject of this paper, the Airborne Hyper-Angular Rainbow Polarimeter (AirHARP, Martins et al. 2019), have demonstrated improved sampling schemes, resolution, and accuracy (Knobelspiesse et al. 2019, van Harten et al. 2018).

Not all polarimetric measurements will achieve a high quality retrieval of cloud DSDs. Multi-angle sampling at high angular density and moderate pixel resolution are essential elements of a confident retrieval. Figure 1 shows theoretical Mie simulations that mimic the polarized cloudbow for particular values of CDR, CDV, and wavelength. To resolve the cloudbow patterns from space and retrieve the CDR and CDV of the cloud, the multi-angle polarimetric instruments must satisfy a minimum viewing angle density (Miller et al. 2018), which is directly related to scattering angle coverage. The *location* of the supernumerary peaks in scattering angle encode CDR, in the 0.67um example shown in Fig. 1. To resolve the CDV, the *amplitude* of the supernumerary peaks must be detected. Polarimeters with coarser viewing angle separation for a single wavelength (i.e. > 3º at 0.67 µm for typical droplets < 15 µm, Miller et al. 2018) may not distinguish these cloudbow oscillations. Instruments like POLDER, which samples at 14 unique viewing angles separated by 10°, do not provide enough native angular resolution (Shang et


al. 2015), and as a consequence, may not be able to identify wide versus narrow DSD clouds at specific geometries (Miller et al. 2018). Only when sampling all native 6x7km POLDER pixels inside a 150km superpixel, they can access the full scattering angle coverage in Fig. 1 and perform a confident retrieval (Breon and Goloub 1998). However, this limits their retrievals to large-scale, homogenous marine

stratocumulus clouds with narrow DSDs. In a study by Breon and Doutriaux-Boucher (2005), a comparison between POLDER polarized and MODIS radiometric retrievals showed a CDR bias of 2 μm that could not be fully decoupled from the large POLDER superpixel. Later evaluation by Alexandrov et al. (2015), with the RSP and the Autonomous Modular Sensor spectrometer, found that the CDR values retrieved by the two methods agree at narrower resolution. Shang et al. (2015) improved the POLDER

retrieval by reducing the superpixel to 42 km. Even though sampling at higher resolution produced gaps in cloudbow coverage, they still found heterogeneity inside the original 150 km superpixel using this improved method. In a follow-up paper, Shang et al. (2019) showed that the POLDER retrieval is sensitive to a wider CDR and CDV range and can be done at a lower 40-60 km resolution when considering all three polarized wavelengths (490, 670, and 865nm) in the retrieval. Even so, no instrument thus far has

performed a polarimetric cloud retrieval from space with both co-located pixel resolution less than 40km and high native angular density (<10°). These goals are essential to study the spatial distribution of DSDs for heterogeneous, broken, and popcorn cumulus cloud scenes, other than the conventional retrievals from marine stratocumulus cases.

The benefit of aircraft instruments like RSP, AirMSPI, and AirHARP is to demonstrate new

technologies that improve upon the POLDER retrieval heritage. RSP, in particular, samples at 150+ viewing angles, separated on average by ~0.8°, and does so for a co-located 250m along-track pixel (Alexandrov et al. 2012b, 2015, 2016). This advancement removes any large scale homogeneity assumptions and allows for a *rainbow fourier transform* on the data, one that retrieves the DSD itself, including multiple modes, without any assumptions on the distribution shape (Alexandrov et al. 2012b).

RSP can sample other kinds of clouds, including broken and popcorn cumulus clouds, with high angular and spatial resolution and does so with high polarimetric accuracy (Cairns et al. 1999). The single pixel cross-track swath of RSP, however, restricts its spatial coverage: RSP cannot form an intuitive image of the scene, requires specific solar angles for cloudbow coverage (Alexandrov et al. 2012b), and requires input from other co-incident instruments for off-nadir context (Alexandrov et al. 2016a). Conversely, the

AirMSPI instrument is a highly accurate pushbroom imager capable of discrete, programmable viewing angles on the same target, but has the same angular limitations as POLDER in this *step-and-stare* mode. AirMSPI also samples in a *continuous sweep* mode that trades co-located information for scattering angle coverage (Diner et al. 2013). This mode gives full visual coverage on the cloudbow, but limits their retrieval to a line-cut of binned pixels along the solar principal plane. A study by Xu et al. (2018) extended

the AirMSPI line-cut retrieval to the entire *continuous sweep* image of the cloudbow with assistance from modeled correlations between COT, CDR, and CDV. This line-cut polarimetric technique requires a droplet size homogeneity assumption over the full line-cut of the cloudbow, which may blur heterogeneity that exists at the pixel-level and steer the retrieval towards wider DSDs.

There is a strong interest in the Earth science community for a multi-angle polarimeter concept

for aerosol and cloud retrievals with a wide swath for spatial context, high accuracy in polarization, high angular density for cloudbow retrieval, and narrow ground resolution (Remer et al. 2019, Dubovik et al. 2019). The Earth and Space Institute (ESI) at the University of Maryland, Baltimore County (UMBC)





designed, developed, and deployed the Airborne Hyper-Angular Rainbow Polarimeter (AirHARP), a next generation wide-field of view (FOV) imaging polarimeter specifically for this purpose. AirHARP is the aircraft demonstration of spaceborne technology that will fly on a standalone CubeSat platform in 2019 in the orbit of the International Space Station, and an enhanced HARP sensor for the NASA Plankton-Aerosol-Cloud-ocean Ecosystem (PACE) mission, called HARP2, in the early 2020s.

In this paper, we will first describe the HARP concept, with a focus on the AirHARP instrument and its data as a proxy for upcoming HARP CubeSat and HARP2 space instruments. We then explain the cloud droplet retrieval framework in Section 3, followed by applications of the retrieval on a stratocumulus cloud deck observed by AirHARP during the NASA Lake Michigan Ozone Study field campaign in 2017, in Section 4. In Section 5, we make use of the fine spatial resolution of the retrieved DSD parameters to explore the information content of the retrieval itself and relate the spatial variability of the results to cloud processes. Section 6 discusses the uncertainties and current limitations of the procedure and we conclude the paper in Section 7, looking ahead to HARP CubeSat and HARP2 deployment and data content.

## 2 Airborne Hyper-Angular Rainbow Polarimeter

The AirHARP concept, and HARP family of polarimeters in general, was developed with a wide swath, fine angular resolution, and high polarization accuracy to address some of the limitations of modern polarimeters. The three HARP instruments, shown in Figure 2, are amplitude-splitting, wide FOV polarimetric cameras. Incident light enters through the wide FOV front lens, passes through telecentric optics, and is split by a Phillips prism toward three detectors. Before reaching each detector, this light passes through a polarizer, oriented at 0, 45, or 90º. The polarizers are oriented at 45º separations such that the I,Q, and U Stokes parameters of the scene can be retrieved in a single co-aligned pixel from an orthogonal basis set of polarization states (Borda et al. 2009). AirHARP images a ground scene with a ±57° (±47°) along-track (cross-track) FOV, and a custom stripe filter over the detector assigns 120 along-track portions, or *view sectors*, of the FOV to four visible channels (bandwidths): 440 (14), 550 (12), 670 (18), and 870 (37) nm. A view sector specifically defines a segment of the detector that corresponds to a unique average viewing angle at the front lens. The hyper-angular 670nm band samples at 60 view sectors, at an average 2° separation, and the other three channels sample at 20 each, at an average 6° separation. In this way, the 60 670nm view sectors can sample the cloudbow oscillations at high angular density without large-scale homogeneity assumptions or degrading the measurement for scattering angle coverage. The wide FOV also allows for broad scattering angle coverage from space during the daylit portion of an orbit. The 20 view sectors in the other three channels ensure multi-angle coverage on aerosols; several studies show that less than seven unique views in a single channel are appropriate for high accuracy retrieval of aerosol optical properties (Hasekamp and Landgraf 2007, Hasekamp 2019), though the details are beyond the scope of this paper. AirHARP is a pushbroom imager, meaning that consecutive measurements from a single view sector can be stitched together to form an image of a ground scene as observed only from that angle. These pushbrooms can have any along-track length, but a cross-track swath proportional to the flight altitude multiplied by a factor of 2.14. This factor accounts for the



maximum AirHARP cross-track view angle, ±47º. A unique pushbroom is made for each of the 120 view sectors, and post-processing registers all of them to a common grid.

A target, either on the ground or in the atmosphere, will be viewed from a subset of the 120 view sectors with its reflected apparent I, Q, and U measured in each view sector and wavelength. From these measurements, the polarized reflectance as a function of scattering angle can be compared with theoretical Mie calculations, as in Fig. 1. The hyper-angular capability of the 670nm channel with its 2° viewing angle resolution can best measure the supernumerary location and amplitude of the cloudbow structure, and therefore, is best for retrieving CDR and CDV of the target cloud. Note that because AirHARP is an imager, each pixel in the image is a potential target viewed by multiple angles. Therefore, each pixel in the image can produce its own polarized reflectance, and can be used to retrieve CDR and CDV, granted that the range of view angles spans a sufficient range of scattering angles. Note that scattering angle range is dependent on both view angle range (fixed by the instrument) and solar geometry (not fixed). If a large number of pixels in the image are viewed at the correct geometry then a spatial map can be made of the DSD parameters across and along the swath, wherever a cloud pixel is found. Depending on the observation altitude and binning scheme, ~0.2 to 6km native retrieval resolutions are possible. Therefore, the microphysics of individual fair weather cumulus clouds can be retrieved across a cloud field stretching tens to hundreds of kilometres. This capability is unprecedented of any existing multi-angle polarimeter instrument.

In HARP's current configuration, all of this retrieval potential fits entirely inside a 10x10x15cm enclosure. The flagship version of HARP is a spaceborne CubeSat, a standalone payload funded by NASA's Earth Science Technology Office (ESTO) In-space Validation of Earth Science Technologies (InVEST), and in collaboration with the Space Dynamics Laboratory (SDL) in Logan, Utah, USA. The HARP CubeSat satellite will be launched in 2019 to the International Space Station orbit (400km, 51.6° inclination) and then dispatched from the station for an autonomous year-long mission. HARP CubeSat will perform cloud retrievals at a minimum 4km superpixel, a capability demonstrated in this paper using AirHARP, a near-identical copy of the CubeSat instrument for aircraft. A third HARP concept, HARP2, is currently under development for the PACE mission to launch in the early 2020s. The HARP CubeSat will be the first satellite to perform wide swath polarized cloud retrievals at sub-5km co-located resolution from space, and HARP2 will continue this capability forward and expand it to provide global coverage in two days.

The remainder of this paper will discuss the information content retrieved from complex cloud scenes observed by AirHARP. The study below refers specifically to AirHARP datasets, but the HARP term may be used when discussing general performance expected from any of the HARP instruments.

## 3 Retrieval framework

A simple treatment of the parametric retrieval is described below, with main components derived from Breon and Goloub (1998), Alexandrov et al. (2015) and Diner et al. (2013) studies. The interaction between incident light and a liquid water cloud droplet is described by a scattering matrix:


$$\begin{bmatrix} I \\ Q \\ U \end{bmatrix}_{sca} = \frac{\sigma_{sca}}{4\pi R^2} \begin{bmatrix} P_{11} & P_{12} & 0 \\ P_{12} & P_{22} & 0 \\ 0 & 0 & P_{33} \end{bmatrix} \begin{bmatrix} I \\ Q \\ U \end{bmatrix}_{inc}, \tag{1}$$

where a Stokes column vector describes the incident beam (subscript *inc*), in total radiance (*I*) and polarized radiance (*Q,U*), and the scattered beam by a similar vector with subscript *sca*. In general, 16
elements describe the scattering matrix but since circular polarization in the atmosphere is negligible (Cronin and Marshall 2011) and not measured by AirHARP, the fourth column and row are neglected. Cloud top liquid water droplets are spherical, randomly oriented, and mirror-symmetric: any matrix elements in Eq. (1) that describe asymmetry are neglected and the others mirror across the main diagonal (Hansen and Travis 1974). The unitless $P_{mn}$ matrix elements scale by the droplet scattering cross-section
($\sigma_{sca}$) weighted by the inverse of droplet surface area.

Sunlight incident on the atmosphere is unpolarized ($Q_{inc}$, $U_{inc} = 0$). For single-scattered photons, the scattered intensity ($I_{sca}$) is proportional to the first matrix element, $P_{11}$, and its polarization ($Q_{sca}$) to the second, $P_{12}$, called the *polarized phase function*. $U_{sca}$ does not contain any structural information in the scattering plane, though it may show a weak linear slope in the presence of non-cloud scatterers
(Alexandrov et al. 2012a). For this reason, $Q_{sca}$ in the scattering plane represents the entire polarized signal.

At the top of the atmosphere (TOA), remote sensors do not observe the scattering from individual droplets but the bulk behavior of the droplet distributions due to measurement resolution and scale limitations. The bulk Mie polarized phase function, $<P_{12}>$, is a weighed sum of optical properties, below:

$$\langle P_{12}(\lambda, \theta, CDR, CDV) \rangle = \frac{\sum_i P_{12,i}(\lambda, \theta, CDR, CDV)\omega_i(\lambda)C_{ext,i}(\lambda)}{\sum_i \omega_i(\lambda)C_{ext,i}(\lambda)}, \tag{2}$$

where $\omega$ is the single scattering albedo (SSA, 1 for water droplets), and $C_{ext}$ is the scattering cross section, which itself is composed of the scattering efficiency and a size distribution weighted by droplet cross-
section. This study uses the same unimodal gamma size distribution function as Breon and Goloub (1998). Polarized reflectance observed at TOA from liquid water cloud droplets is proportional to $P_{12}$, after a correction for viewing geometry:

$$\mathcal{R}_{obs} = \frac{4}{\pi}(\mu_0 + \mu)\left[\frac{-\pi Q_{sca}}{\mu_0 F_0}\right], \tag{3}$$

where the cosines of the view zenith angle ($\mu_0$) and solar zenith angle ($\mu$) and the band-weighted extraterrestrial solar irradiance ($F_0$) rescale the polarized radiance ($Q_{sca}$). The bracketed term is the *polarized reflectance* ($\rho_P$), and a similar expression gives the total reflectance ($\rho$) using the Stokes parameter $I_{sca}$ in place of $Q_{sca}$. Subsequent Figures use $L_{670nm}$ for $I_{sca}$ and $L_{P,670nm}$ for $Q_{sca}$ radiances, where
applicable, and anytime the term *intensity* is used, it corresponds to a radiance measurement, not reflectance, unless explicitly noted. Because we are only using a single wavelength in our retrieval, radiance and reflectance are interchangable, in terms of information content shown in the Figures.



Corrections to Eq. (3) for Rayleigh scattering at observation height are performed in prior studies (Breon and Goloub 1998, Diner et al. 2013), but the necessity is disputed (Alexandrov et al. 2015): this study accounts for Rayleigh effects in a weak cosine term described below.

The retrieval compares the Eq. (3) to a parametric model and infers the CDR and CDV from the best fitting $P_{12}$ simulations:

$$\mathcal{R}_{fit}(\lambda, \vartheta_{scat}) = \alpha P_{12}(\lambda, \vartheta_{scat}, CDR, CDV) + \beta \cos^2 \vartheta_{scat} + \gamma. \qquad (4)$$

     The parametric fit scales the theory, Eq. (4), to observations, Eq. (3), inside the polarized
cloudbow scattering angle range ($135° > \vartheta_{scat} > 165°$, Di Noia et al. 2019, Shang et al. 2015) with three free parameters ($\alpha$, $\beta$, $\gamma$). Corrective factors for aerosol above cloud, cirrus, sun glint, molecular scattering, and surface reflectance signals comprise weak functions of scattering angle, with the parameter $\alpha$ related to cloud fraction (Breon and Goloub 1998, Diner et al. 2013, Alexandrov et al. 2015).

     A prescribed look-up table (LUT) in CDR and CDV drives the parametric fit, ranging between 5
and 20µm in CDR ($\Delta = 0.5$µm), and CDV values of 0.004 to 0.3 at variable intervals, similar to Alexandrov et al. (2015), with $\Delta$ values indicating the step size. The LUT is dense for CDV < 0.1: the majority of supernumerary bow sensitivity exists below this level and reduces considerably for CDV > 0.1, as shown in Fig (1). Polarized reflectance measurements are corrected via Eq. (3) and fit in a non-linear least-squares process to Eq. (4), checking all possible combinations of CDR and CDV in the LUT.
The root-mean-square error (RMSE) and reduced chi-square statistic $\chi^2_{red}$ of the least-squares process verify all LUT comparisons:

$$RMSE = \sqrt{\frac{1}{n} \sum_i^n \left( \mathcal{R}_{fit,i} - \mathcal{R}_{obs,i} \right)^2}, \qquad (5)$$

  and

$$\chi^2_{red} = \frac{1}{n-5} \sum_i^n \frac{(\mathcal{R}_{fit,i} - \mathcal{R}_{obs,i})^2}{\sigma_{obs,i}^{\;2}}. \qquad (6)$$

     The $\chi^2_{red}$ verifies that the data is best described by the fit in Eq. (4) with $n$-5 degrees of freedom (for 3 fit parameters, CDR, and CDV), where $n$ is the number of measurements in the cloudbow scattering angle range for that pixel. Like Alexandrov et al. (2015), a fine scale interpolation is performed on the
30 LUT at 10 times the original resolution in CDR and CDV. Retrievals are accepted immediately for $\chi^2_{red}$ values 0.5 to 1.5. In this range, our error estimate is consistent with the minimized fit. If the $\chi^2_{red}$ is outside this range, our error may lead to an overfit ($\chi^2_{red} < 0.5$) or underfit ($\chi^2_{red} < 1.5$). However, large $\chi^2_{red}$ does not always mean the fit is poor in our case: the physics of the cloud field may justify solutions with $\chi^2_{red}$ beyond 1.5. Therefore, we also check to see if the fit satisfies an RMSE threshold of 0.03. If
not, the fit is rejected and the pixel is flagged. These diagnostics were found by a sensitivity study on synthetic AirHARP cloudbow retrievals and an estimation of the error in the actual data.

     There are several reasons for this two-factor authentication. First, we recognize the signal-to-noise of the superpixel is not the only error that contributes to the measurement. Optical etaloning that remains



in this AirHARP dataset will also add uncertainty. This effect is weak compared to the signal and nearly random angle-to-angle, so we estimate an extra 1-sigma contribution in each superpixel to account for it. Therefore, the superpixel uncertainty, $\sigma_{obs}$ used in Eq.(6) represents two times the standard deivation of the superpixel bin. Because the $\chi^2_{red}$ value depends heavily on a correct error estimation, it is important

that all artifacts in the data are well-accounted for. Second, there is evidence in the literature that when multiple DSDs exist inside the same superpixel, the polarized signal will not agree completely with a signal that represents a single DSD (Shang et al. 2015). This retrieval will still attempt to find a representative DSD in the measurement, however. Here, the $\chi^2_{red}$ may be higher than 1.5, but the RMSE threshold can still find a solution if the measurement residuals are not too far from the best fit curve. This

may also occur for observations of multi-layer cloud fields. Third, Breon and Goloub (1998) noted that secondary and tertiary scattering events in the primary bow region (137-145° in scattering angle) can widen the polarized signal here relative to Mie simulations. Here, the RMSE may preserve a strong fit in the supernumerary region, where the majority of the DSD information content lies, even if the $\chi^2_{red}$ is beyond our threshold. These diagnostics also account for artifacts from rotating into the scattering plane

on $Q_{\mathrm{sca}}$ and $U_{\mathrm{sca}}$ and retrievals that poorly converge to CDR and CDV values at the very edge of the LUT. When the uncertainty is high relative to the meausrement, both $\chi^2_{red}$ and RMSE will also be high and the retrieval will be rejected if both values exceed their expected ranges. More details on some of these effects are discussed in Section 6.

The focus of this paper is on the application of AirHARP cloud datasets and not the retrieval
algorithm itself; therefore, we use a simple treatment of the classic parametric model. This retrieval will be extended to multi-modal DSDs and take into account both multi-angle and multi-spectral sampling in future studies.

## 4 Hyper-angular polarized cloud retrievals from AirHARP

Before we discuss how the retrieval is applied to the AirHARP data, we will first walk-through
an AirHARP measurement. As the AirHARP instrument images a scene for a particular solar geometry, each view sector captures a range of scattering angles, unique to each of the 120 view sectors and wavelengths. Figure 3 shows an example of the AirHARP instantaneous scattering angle coverage for a simulated observation at 15:22 UTC over Lake Michigan on 19 June 2017, during the NASA Lake Michigan Ozone Study (LMOS) field campaign. This target was chosen because of the cloud conditions present

during the observation and the solar geometry allows for retrievals across the swath and along the entire length of the observation. Figure 3 shows a simulated cloudbow as it would appear in a single AirHARP snapshot, if the entire detector was capable of sampling at 670nm. This cloud field was simulated using a CDR of 10um and CDV of 0.01, with the same solar and viewing geometry of the LMOS observation. Note that this is the scattering angle coverage for a single snapshot, and when AirHARP flies over a cloud

deck, it is taking two snapshots a second. This means a different portion of the detector is imaging the same cloud target from image-to-image, which also suggests the scattering angle observed at the target changes image-to-image as well. From the perspective of the detector, the target travels from the front of the detector to the back during a full angle observation, reflecting solar light at different scattering angles as the instrument flies over it. Therefore, only along-track pixel columns inside the yellow-dotted lines in





Figure 3 contain pixels that are eligible for a polarimetric DSD retrieval. Targets observed outside these lines do not access the full cloudbow scattering angle range (<165°).

Figure 4 shows the view sector isolines of AirHARP over the same snapshot from Figure 3. The AirHARP wide FOV covers view sectors from ±57°, but note that the cloudbow only covers a subset of these. Pushbrooms are made from individual view sectors as the instrument flies over the cloud field. Figure 4b shows examples of pushbrooms built from cloudbow content in Figure 4a isolines. If AirHARP was to fly over this simulated field, the cloudbow would transition from a concentric space in the raw image to a linear one in the pushbrooms. This occurs because each view sector only observes a specific cross-section of the cloudbow at any one time, and the structure of the cross-section is maintained due to the geometry of a single view sector. Figure 5 shows the actual AirHARP observation at view sectors near +38º during the time and day used to simulate Figures 3 and 4, during the LMOS campaign. The actual image displays the cross track cloudbow structure of the segment near +38º in the Figure 4b simulation. The RGB composite shows off the wavelength dependence in the polarized cloudbow structure, which is absent from the total reflectance image. Also, the appearance of the cloudbow in this pushbroom is highly variable compared to the simulation in Figures 3 and 4, which reflects the heterogeneity in the cloud field seen in total reflectance.

Since a single target moves across the detector in consecutive snapshots, there will always be a location in each of the 120 pushbrooms that represents that target on the ground, and any cloudbow target appearing in multiple sector views having sufficient scattering angle range can be used in a polarimetric cloud DSD retrieval. Figure 6 shows several examples of an AirHARP 200m superpixel retrieval of different regions of the LMOS cloud field shown in Figure 5, using hyper-angular, co-located information. Error bars represent two times the standard deviation of the pixels inside the superpixel bin. Superpixels are constructed from finer resolution native pixels to increase SNR and mitigate other potential artifacts in the data. These artifacts will be discussed in Section 6. Note that Figure 6a and 6b both represent narrow DSDs, with low CDV values, though the difference in CDR forces a shift to the location of the observed supernumerary bows. Figures 6c and 6d are wider DSDs, with higher CDV values, with eroded supernumeraries. As the DSDs become wider and wider, this retrieval method becomes less and less accurate at inferring CDV, as the supernumerary region becomes monotonic and linear. The CDR values retrieved in Figure 6 are typical of non-precipitating stratocumulus cloud fields (Pawlowska et al. 2006) and CDV values are similar to those found by Alexandrov et al. (2015) using RSP measurements over marine stratocumulus.

The hyper-angular retrieval requires data that is captured over a short time window as AirHARP flies over a cloud: it takes time for the AirHARP backward angles to image the same location on the ground as the forward angles. The differences in time depends on the instrument-level flight speed and difference in altitude between the instrument and target. For the LMOS campaign, the difference between ±57° observations was 112 seconds (~2 minutes) for nominal UC-12 flight speed of 133 m/s at 4.85km altitude above the cloud deck. Note that actual above ground altitude was 8 km, but the cloud deck was geolocated to be 3150m on average). Therefore, the hyper-angular retrieval requires cloud constancy over this time interval. If we only include the angles used in the cloudbow retrieval, the time interval between the views with the largest angular separation reduces to a minute. A study with the HARP CubeSat at estimated 400 km orbital altitude and 7.66 km/s ISS speed requires ~160 seconds (~2.5 minutes) for the





same full-angular coverage over the same cloud target. In this way, the HARP hyper-angular retrieval still requires an assumption of homogeneity in a short time window over a narrow pixel.

With this in mind, any liquid water cloud pixel in the AirHARP wide FOV that samples scattering angles between 135-165° can be used to retrieve CDR and CDV. This constraint is used in several other
polarimetric studies, though with a slight discrepancy on the start of the lower bound (Di Noia et al. 2019, Alexandrov et al. 2015). Shang et al. (2015) found that using 137-165° scattering angle range as opposed to the operational POLDER 145-165° improved many of the CDR and CDV retrievals, specifically for CDR > 15μm (Shang et al. 2019). The upper bound of 165° is consistent between studies dating back to Breon and Goloub (1998): the bulk of the microphysical information lies in the supernumerary bows and
the assumption of a structureless $U_{sca}$ breaks down after this point (Alexandrov et al. 2012a). Figure 7 shows an example of how individual pixel retrievals generate a spatial distribution of CDR and CDV, for those that access this cloudbow scattering angle range. Each pixel is first conservatively masked for non-clouds using the nadir 670nm intensity pushbroom (-0.003° VZA) using a conservative threshold of 0.06 W m$^{-2}$ sr$^{-1}$ μm$^{-1}$ to avoid cloud holes and views of Lake Michigan below. All pixels are aggregated to 4x4
resolution (200m) and the polarized radiances ($L_{P,670nm}$) are converted into polarized reflectances, via Eq. (3), before entering the retrieval process.The portion of the image capable of retrieval stretches 34km along-track and 3km cross-track.

The distribution of CDR and CDV in AirHARP data is consistent with prior studies and physical phenomena. Because the cloud case observed during LMOS was heterogenous, there are several examples
of how cloud substructure can give different retrievals. Figure 8 takes a few areas from Figure 7 and zooms in on their retrieval results. Figure 8b shows a uniform sector of the cloud field, described this way because of its visual homogeneity in both intensity and CDR and the narrow and consistent CDV retrievals over many pixels. The results here suggest the supernumerary bows are well-defined and the cloud pixels have narrow size distributions. Figure 8a shows a region of the same leg that is heterogeneous
in CDV, and the intensity and CDR distribution suggest that this area is a region of convection: larger CDR in the cloud core, or central area of the cloud, and smaller CDR retrieved on the periphery, where the intensity is lower. We will look at this phenomenon in more detail in the AirHARP data in the Sections below. Here we point out that large-eddy simulations (LES) of similar heterogenous clouds show similar spatial distributions of intensity, CDR and CDV (Miller et al., 2018), with one representative case shown
in Figure 8c. Miller et al. (2018) simulate LES clouds using vertical weighting functions that take into account the distribution of reflectance at the edges of the cloud, echoing theoretical recommendations made by Platnick et al. (2000).

While these simulations can assume any resolution, the AirHARP retrievals are performed at 200m in this study and even coarser resolutions from space. The small-scale variability in the cloud field
can also be missed by MODIS radiometric analyses, for example, which assumes constant CDV in their droplet size retrieval. This is one of the strongest benefits of polarized cloud retrievals: a quantitative measurement of heterogeneity through CDV information, which is not possible with traditional radiometric methods. This has serious implications to climate in terms of quantifying cloud development, brightness, and lifetime, aerosol-cloud interaction, and reducing the uncertainty in global radiative forcing
due to clouds and aerosols. In the following section, we explore how we can extend the AirHARP spatial retrievals of CDR and CDV to study changes in size properties along the cloud field and the impact of resolution on the retrieval itself.



## 5 Spatial scale analysis

Because the AirHARP retrievals of CDR and CDV are images, any sector of the cloud field can be analysed by taking a transect of pixels along- or cross-track. In Figure 9, we take a 34km pixel transect
of the cloud field (shown in the inset intensity image with a black line), and compare the anomaly from the mean along the track for intensity, CDR, and CDV. The CDV is log-scaled to linearize its several orders of magnitude range. Positive CDR anomalies describe larger droplet sizes, and positive CDV anomalies correspond to wider distributions. Any position along the transect of the cloud field lines up exactly with three unique points in the plot, and the correlation between the three curves suggests infor-
mation about the nature of the cloud field. It is important to note that we are using the cloud intensity as a proxy for COT, which is orthogonal to CDR (Nakajima and King 1990). In some locations in the plot, intensity (COT) and CDR are correlated with each other while anti-correlated with CDV. Blue blocks define unambiguous locations in the cloud field where intensity and CDR have positive anomalies while the CDV anomaly is negative, whereas orange blocks give the opposite: intensity and CDR are negative
and CDV is positive. If we define cloud cores as the pixels brightest in intensity (blue) and cloud periph-eries as darkest in intensity (orange) then cloud cell sizes appears to be on the order 1-4km, both compa-rable to and slightly larger than traditional MODIS cloud droplet size retrieval products (1 km). Compar-ison to the traditional cloud product resolution is notable because, in some regards, the 1 km resolution is adequate to resolve cloud microphysics of the cloud cores. However, when cores and peripheries are
found in the same 1 km pixel, issues in separating DSDs will arise. Since AirHARP is an aircraft instru-ment and flies beneath 20km, its resolution will be better than an equivalent AirHARP instrument in space, so likely this fine-scale variability will not be captured by HARP CubeSat or HARP2. Regardless, this result emphasizes the importance of small-scale sub-kilometer sampling of cloud fields because cloud heterogeneity and microphysical processes may be lost in the large spatial resolutions of spaceborne in-
struments.

There are physical explanations for the relationships we see between intensity, CDR and CDV on the spatial scales of Figure 9. Liquid water droplets that form at the base of adiabatic clouds, such as cumulus and stratocumulus, see their largest sizes at cloud top (Platnick et al. 2000), and further grow by longwave radiative cooling, small-scale turbulence, and collisional processes (de Lozar and Muessle
2016). On the periphery, evaporation removes smaller droplets, and at the same time, entrainment of warm air and/or aerosol here may enhance droplet growth. There are many competing theories as to the net effect of aerosol entrainment on droplet growth (Small et al. 2009 and references therein), but these two opposing effects may create a larger DSD variance on the periphery. Alexandrov et al. (2015) and Platnick et al. (2000) suggest that CDR changes occur vertically in the cloud periphery. Therefore, multi-
angle polarimeters that can sample deeper into the periphery could retrieve a larger CDV in these areas. The above LES study of broken marine stratocumulus by Miller et al. (2018) also show higher CDV (lower CDR) in cloud periphery and lower CDV (higher CDR) in cloud cores, as shown in Figure 8c. In the present study, all of these processes cannot be decoupled, but these promising results show that AirHARP retrievals are consistent with current research and theories of cloud microphysics.



Furthermore, the AirHARP pixel resolution can be degraded and used to understand the effect of sub-pixel variability on the DSD retrieval itself, as shown in Figure 10. Figure 10a and 10b are repeated from Figure 7a and 7b, and both represent the 200m CDR retrieval, while Figure 10c shows the CDR product at 600 m resolution. To calculate the 600 m product, the gridded polarized reflectance data at the original 50m resolution are aggregated into 600 m superpixels. Second, the superpixels pass through a screening process: this eliminates low intensity superpixels that represent cloud holes and marginal situations. Third, the superpixels enter the retrieval process. Thus, Figure 10c is not a resampling of Figure 10b, but a new retrieval using a different resolution as input. This study does not examine the effect of cloud screening at the different resolutions, only the effect of the degraded resolution using pixels that have been properly identified as clouds at the finer resolution.

The plots shown on the left-hand side of Figure 10d-f are the retrieved $P_{12}$ curves, which emphasize how the nine 200 m retrievals, in the gray lines, compare to the single 600m retrieval, the red line. Figure 10d shows that the narrow DSD retrievals are robust against resolution degradation; if we take the 200 m retrievals as truth, the 600 m result agrees within community standards (10% $\sigma_{CDR}$ and 50% $\sigma_{CDV}$, Mischenko et al. 2004). The 600 m $P_{12}$ resembles the 200 m $P_{12}$ curves, both in location of supernumerary peaks and overall structure.

Figure 10e shows a retrieval that appears to represent the cloud periphery, as the intensity image shows the appearance of a cloud cell near the superpixel. Here, the CDR retrieval gives higher values in the center of the structure and smaller values on the sides, consistent with prior studies. Figure 10e shows two conflicting $P_{12}$ regimes. Here, 200 m DSDs with CDR between 6.6 and 7.5um separate into two modes: CDV >0.08 and CDV between 0.048 and 0.028. While the primary bow around 143° is preserved between retrieval scales, the 600 m retrieval gives a CDV of 0.047, a value that appears to represent the mean of the nine pixels, but satisfies neither regime. Shang et al. (2015) and Miller et al. (2018) show similar results in theoretical and observational mixed DSDs. Note the combination of gamma distributions inside a superpixel is not itself a gamma distribution, though retrievals that contain sub-pixel heterogeneity in the DSD still attempt to infer gamma distribution properties from a signal that may not represent one (Shang et al. 2015). The *rainbow fourier transform* method (Alexandrov et al. 2012b) may distinguish these two modes at the 600 m scale, but the result could not be independently validated if it was performed with RSP single-pixel sampling, as it is here with AirHARP data. Figure 10f shows another retrieval done close to the cloud periphery, but this time, the retrieved 200 m $P_{12}$ curves are well-mixed in CDV. The retrieved 600 m fit generates a curve that does not represent any of the subpixel results. The consequence is a broad 600 m CDV that reflects the 200 m variability, but not the mean magnitude of the 9 subpixels, as the 600 m retrieved value for CDV is 0.284, while the 9 individual pixels return values 0.086 to 0.186. Here, the subpixel variability smears out the supernumerary bows. This result is a well-known consequence of mixed DSDs in a large superpixel, but does not provide any information as to which parts of the cloud inside the superpixel contain narrow vs. wider DSDs. The interpretation of CDR and CDV at large pixel sizes is still widely debated, but fine resolution spatial data provided by AirHARP and its retrievals can provide a meaningful advancement in this direction.



## 6 Discussion of Limitations and Uncertainty

The first limitation of this method concerns the parametric retrieval, which assumes a single mode DSD that can be described by CDR and CDV alone. Situations that do not fit this assumption may not be retrievable, as mentioned above. We note that other retrieval methods will overcome this limitation. However, for the purposes of this initial demonstration, the assumptions of the parametric retrieval seem to be met by our example.

This being said, we cannot ignore the possibility of optically thin upper level clouds ($\tau_{\mathrm{cld}} < 1$) moving over the geolocated cloud deck. If these clouds exist, they will appear to "move" from angle to angle, as our current geolocation algorithm, discussed later on, focuses on the layer of clouds that is producing the dominant signal. This also means their impact on the cloud retrieval will change angle-to-angle, but will likely affect only two or three view sectors at most with a weak contribution to the measurement. Therefore, this is not expected to significantly contribute to the overall cloudbow fit. All fits shown in this manuscript lie beneath our successful RMSE threshold, supporting this claim, though other retrieval methods could tease out the signals from both cloud layers when properly geolocated (Alexandrov et al. 2012b, 2016).

This limitation, the need to map the different angular measurements to a target elevation, is a significant challenge with AirHARP data. If the target were Earth's surface, then a digital elevation map could be used, but clouds appear at a range of altitudes and are not always easily predictable in height, distribution, time, or space. Therefore, an iterative method determines, within a gridded pixel, the altitude that provides zero parallax displacement in the cloud field. We use the location of several distinguished cloud features, as observed from at least two view sectors, to determine the average cloud height of the dataset. Currently, this single cloud height estimate is applied to the entire dataset during the Level 1 geolocation process. Because there is no such thing as a plane-parallel cloud, both the parallax method and assumption of constant cloud-top height introduces uncertainty in the retrieval. Where possible, the altitude of the scene is verified with heights retrieved from data from other co-incident instruments. Conservative cloud identification and binning pixels to 200m (4x4) resolution further mitigates the error introduced by using this mean height. In the case shown in Figure 5, the derived height is (3150 ± 50) meters, where the uncertainty is the resolution that guarantees no movement from view sector to view sector. A self-check on the validity of these assumptions is the goodness of fit of each retrieval. In our example, while the RMSE of the cloud field varies, all retrievals shown successfully fit the RMSE threshold defined above. Therefore, we believe errors in our geolocation do not contribute significantly to the results of our study. However, when studying broken or popcorn cumulus clouds, a proper 3D geolocation of the cloud will be required. The HARP science team is currently developing an optimized pixel-level topographic algorithm to mitigate any multi-layer or cloud projection biases in future retrieval studies, as well as other quality assurance corrections beyond the scope of this work.

During an aircraft campaign, it is also important to maintain calibration accuracy in-flight as many factors such as temperature, pressure, vibration, and humidification can alter the quality of the measurements. AirHARP did not have an in-flight calibration mechanism during LMOS, so it is challenging to verify the accuracy of the in-flight data. We also realized in lab studies that AirHARP generates internal optical etaloning. The etaloning produces concentric fringes on the raw image, which transforms into linear bands at the pushbroom level. Typically, a lab flatfield corrects for this effect, but our flights show


the etaloning occurs in new locations at the detector during flights. The fringes are variable in strength and size across each view sector, but are easily removed over homogenous targets with our current correction scheme. Because the cloudbow also transforms from a concentric to linear space at the pushbroom level, cloudbow cases are especially tricky to correct. Luckily, the effect of the fringes on the hyper-angular retrieval is nearly random: the retrieval is a structural fit from angular data that covers many unique positions in the detector. Therefore, the etaloning can be treated as a decrease in SNR in the measurement, which we estimate as an extra 1-sigma error contribution. Due to the well-resolved retrievals in Figs. (6-10), it is clear that the etaloning does not contribute significantly to the CDR and CDV products, though the fringe contribution would make it more difficult to retrieve above-cloud-aerosol signals hidden inside the cloudbow measurement, if applicable. For this reason, we are currently developing a new correction algorithm to remove the fringing from heterogenous datasets and an internal calibrator for HARP2 on the PACE mission. With frequent flatfield calibrations, the etaloning can be immediately corrected in a variety of environments. Also, there is evidence that the cloudbow retrieval itself could be used to remove the fringe signal from AirHARP data. While the cloud signal changes across multiple full-size images, the fringe structure is stable for the same view sector. This allows for the characterization and removal of the fringes uniquely from cloudbow datasets. This is a promising correction technique that will be explored in future work.

Finally, the results of the retrieval presented in this paper are challenging to validate, and even difficult to compare with other retrievals. The MODIS Terra and Suomi-NPP VIIRS radiometers passed over the same location on the ground as our example over an hour after the AirHARP observation, and while the GOES-R Advanced Baseline Imager (ABI) radiometer is co-incident, its 1-2km CDR retrieval resolution is difficult to reconcile with the variability observed in the cloud field at finer AirHARP resolution, as suggested by our study presented in Figure 10. Intercomparing GOES-R and AirHARP retrievals will be performed in a future study; the constant co-incidence of GOES-R makes it very attractive for field campaigns that once relied on the sparse co-incidence of the polar-orbiting or ISS-based sensors. AirHARP was also the only Earth-observing polarimeter present on aircraft or space during LMOS with these capabilities for cloud retrieval. None of the field experiments in which AirHARP has flown (LMOS or ACEPOL in 2017) focused on clouds, allowing only for a few cloud targets of opportunity during both campaigns. For example, the observation presented in this paper was the only one in which AirHARP achieved full angular coverage over a continuous cloud field, and one that could be geolocated to a constant height over the full pushbroom with little impact to the retrieval itself. A dedicated future aircraft campaign for clouds, specifically one that allows co-incident AirHARP observations with other compatible cloud-measuring or -retrieving sensors at similar spatial resolution would be beneficial for validation. Optimally, co-incident HARP space and AirHARP aircraft observations would greatly improve our ability to validate the differences in retrieval resolution for cloud cores and periphery, across a wide swath, and for unique global locations and aerosol source regions.

## 7 Conclusion

We used the AirHARP hyper-angular measurements at a single wavelength (670nm) in a traditional parameterization scheme to demonstrate the ability of the HARP concept to characterize cloud





microphysical parameters across a cloud field at sub-kilometer spatial resolution. HARP measurements can also be applied to the four polarized wavelengths, akin to Breon and Goloub (1998) and Shang et al. (2019), but this type of retrieval is more sensitive to resolution and calibration than the hyper-angular technique presented here, and most importantly requires homogeneity at cross-track scales (3 km). In the hyper-angular method, we can achieve the same results at the pixel scale (0.2 km) and *resolve* the spatial inhomogeneity across the track. Variability within a 3 km scale retrieval can easily steer the parametric retrieval towards larger CDV (Figure 10e, Shang et al. 2015, Miller et al. 2018) and calibration biases that affect an individual view sector (i.e. etaloning) will be much more prominent and systematic in this approach than in the hyper-angle retrieval. Using data from multiple channels (Shang et al. 2019) serves as excellent cross-calibration and intercomparison with other instruments over narrow DSD marine stratocumulus (Alexandrov et al. 2015, Knobelspiesse et al. 2019), all of which will be explored in future work.

The HARP concept enables highly resolved, hyper-angular polarimetric retrievals of liquid water cloud microphysical properties. Using a heterogenous cloud field from the NASA LMOS campaign as an example, AirHARP datasets allow sub-kilometer, spatial retrievals of CDR and CDV across the full swath and along the entire flight track. These analyses reveal cloud sub-structure and the spatial distribution of DSDs for cloud cores and periphery regions, which can be easily extended to marine stratocumulus, tradewind, and popcorn cumulus clouds. Because of the wide HARP FOV, these retrievals are possible off the instrument nadir scan line and during nearly all daytime hours, if geometry allows. Combined, these capabilities position HARP as an instrument capable of cloud DSD retrieval at scales relevant to climate study and with global coverage from space.

Because of relatively fine spatial resolution retrievals across a broad swath, we were able to perform scale analysis on retrieved DSD parameters by degrading the resolution and sub-sampling the hyper-angular polarimetric retrieval. At the pixel-level, we note that large pixel sizes blur DSD variability, resulting in a retrieval that tries to account for all subpixel regimes but ends in producing an entirely different DSD altogether. The sub-kilometer retrievals were able to identify small-scale DSD variability in our heterogenous cloud field. Specifically, we found that there is a correlation between intensity (a proxy for COT) and CDR, and an anti-correlation with CDV. For places where intensity is high (high COT), assumed to be cloud cores, droplets were large and size distributions narrow. The opposite is found along the cloud periphery. These findings may be related to entrainment and droplet evaporation on cloud edges, and collision-coalescence processes in cloud cores, but further theoretical study and targeted campaigns are needed. Some of these results are not limited to the HARP design: producing spatial images of both DSD parameters can be achieved by other imaging multi-angle polarimeters. Producing these images with the spatial resolution sufficiently fine to illuminate cloud processes along with the spatial coverage to encompass an entire two-dimensional cloud field is unique to the HARP concept.

Future work anticipates extending these concepts to multi-modal size distributions and combining multi-spectral and multi-angle sampling to retrieve cloud size properties and other information: aerosol-above-cloud microphysics, cloud height, and thermodynamic phase, as well as a stronger definition of CDR and CDV for large superpixels that contain internal DSD variability. With the upcoming launch of HARP CubeSat in 2019 and HARP2 in the early 2020s, the same retrieval concepts applied here on Air-HARP data can be used to connect cloud properties to global radiative forcing, improve radiometric



retrievals, and provide strong science rationale for including high-resolution, hyper-angle imaging polar-imetry on future Earth science space missions.

*Data Availability.* Quality-assured AirHARP data from the NASA LMOS campaign to Version 000 can
be found on https://www.air.larc.nasa.gov/cgi-bin/ArcView/lmos. Future version updates are planned as well as delivery of more LMOS datasets to the archive in 2020. Data used in this manuscript and L2 products can be found at https://www.dropbox.com/sh/imd9quoloeqhsum/AADeyvMchZS-rabM8nJ7VEgi-a?dl=0.

*Author contribution.* BM actively participated in the NASA LMOS field campaign: operated AirHARP in the aircraft, performed ground calibrations, and L1 processing of the datasets used in this manuscript. BM wrote the retrieval code, error analysis, and interpretation of the results in this manuscript. LR, JVM, and HMBJ provided substantial editing support, and the latter developed the general algorithm used for AirHARP L1 processing of LMOS datasets. WB was the first to use the algorithm developed by BM to
generate the first spatial CDR and CDV products from this cloud field, which were quality-assured and optimized in this manuscript.

*Competing interests.* The authors declare they have no conflict of interest.

*Acknowledgements.* BM acknowledges funding through NASA NESSF project 18-EARTH18R-40. HMBJ acknowledges funding from FAPESP project 2016/18866-2, and from CNPq project 308682/2017-3. We also thank the LMOS team for hosting the participation of the AirHARP instrument
during their flight campaign.

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

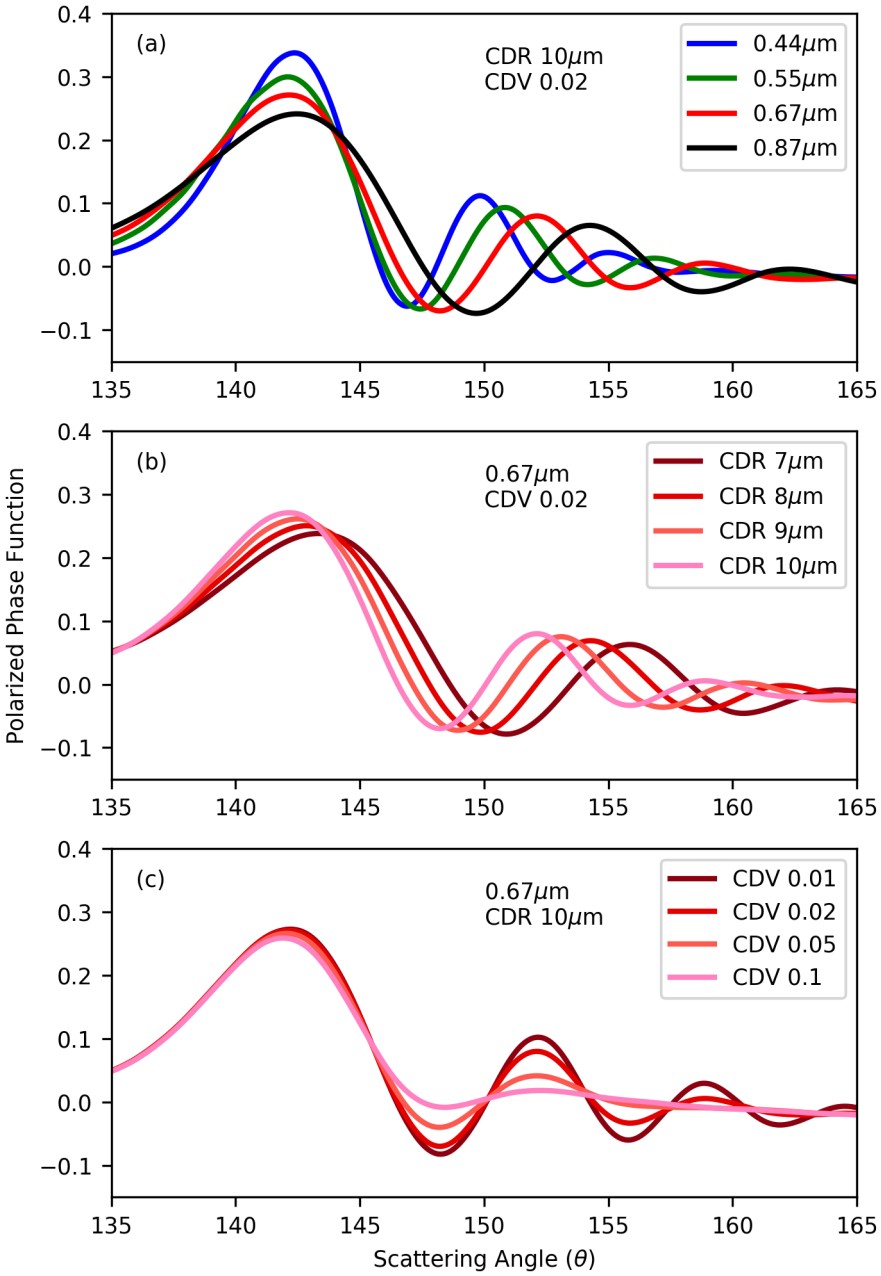

**Figure 1.** *Mie scattering simulations for liquid water cloud droplets, with solar light incident. For the four HARP wavelengths at 10um CDR and 0.02 CDV (a), the 0.67um channel for variable CDR and constant CDV (b) and constant CDR with variable CDV (c).*



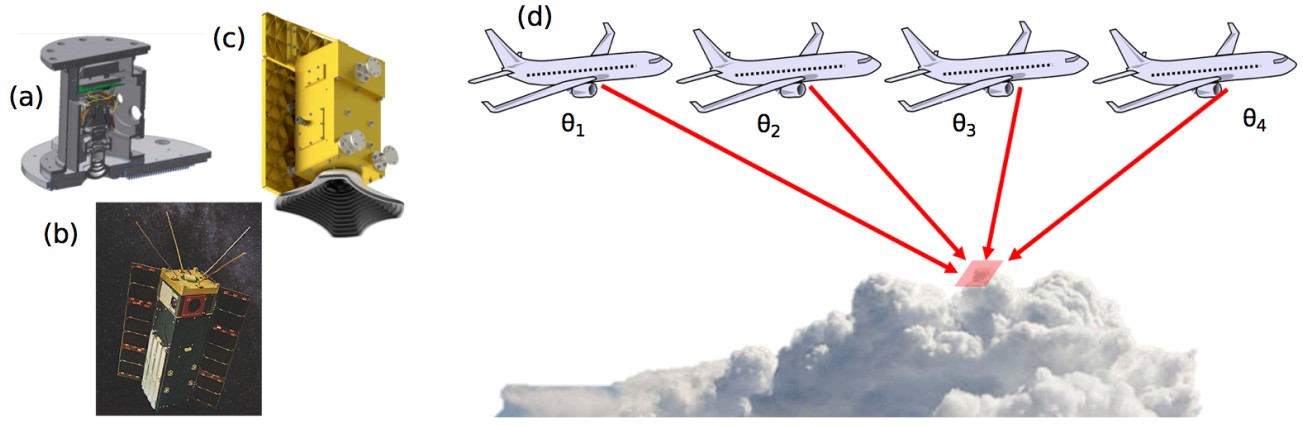

**Figure 2.** *Figures a, b and c illustrate the current HARP VNIR polarimeter family consisting of the Air-HARP airborne system (a), HARP CubeSat (b), and HARP2 for the NASA PACE mission (c). The HARP concept comprises a wide field-of-view imaging polarimeter that images the same ground target from up to 60 distinct viewing angles at 0.67μm (d), and up to 20 viewing angles at 440, 550 and 870nm. The wide cross track swath (94º) of HARP2 allows for global coverage from space within two days.*



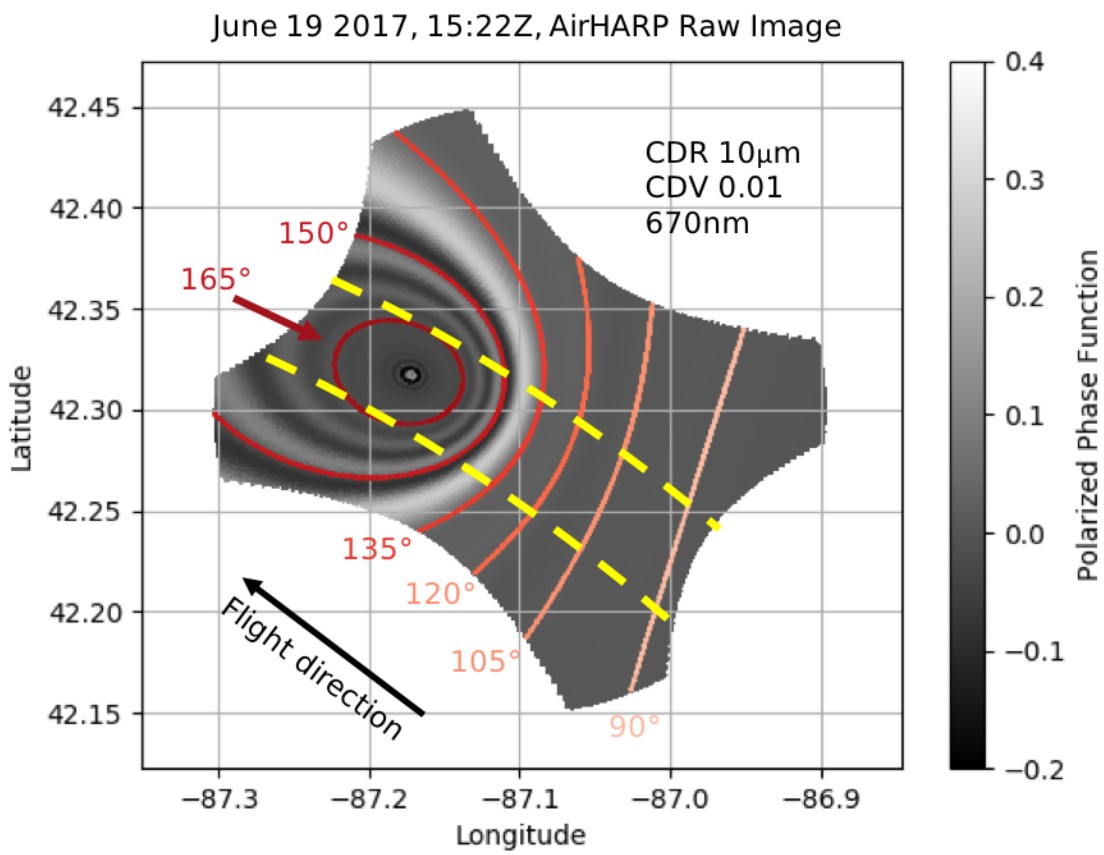

**Figure 3.** *The scattering angle coverage typical of an instantaneous AirHARP wide FOV observation,*
*projected to simulate a scene over Lake Michigan on June 19, 2017 at 15:22Z (a). The cloudbow in this*
*simulation represents a simulated cloudbow at 670nm, with CDR 10um and CDV 0.01, with scattering*
*angle isolines from 90 - 165° shown (solid lines). Note the cloudbow pattern occurs within 135-165° in*
*scattering angle, and the location of the cloudbow in the FOV depends on time of day, flight orientation,*
*and solar geometry. Note that the only portion of the image that is eligible for retrieval lies inside the*
*region defined by along-track lines tangent to the 165° scattering angle isoline (yellow dotted lines).*





**Figure 4.** *The same simulation as Figure 3, now with along-track view sector (zenith angle) isolines (a).*
*A pushbroom is made when a view sector images consecutive information along-track, shown for seven*
*view sectors, separated by 7.5° each (b). Because cross-track pixels represent a cross-section of the*
*cloudbow, they are also proxies for scattering angle. Note that the cloudbow distribution is different for*
*each view sector pushbroom. Since each pushbroom is projected on a common grid, any pixel or*
*superpixel in common to all of the views can generate a discrete polarized structure. This measurement*
*can be compared to Mie simulations to retrieve CDR and CDV at the target resolution.*

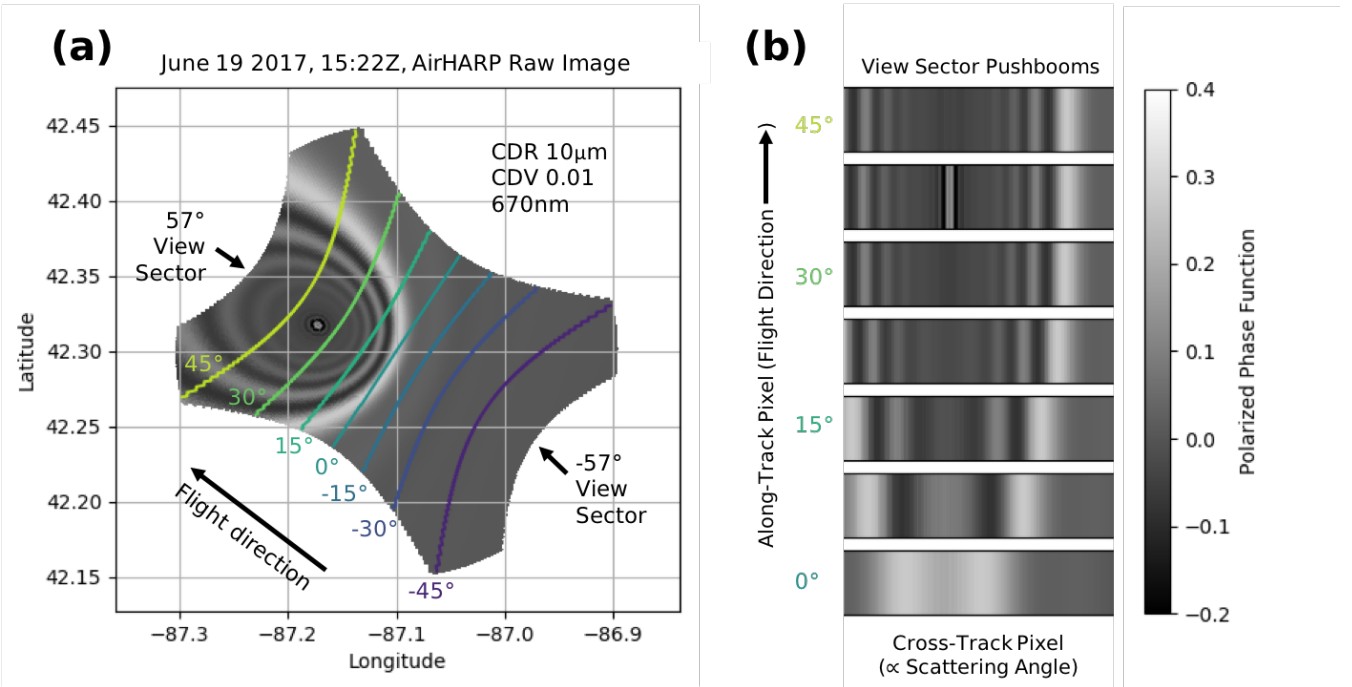





**Figure 5**. *AirHARP data taken at the same time, location, and geometry as Figures 3 and 4 reveals a heterogenous cloud field in total reflectance (a) and a cloudbow in polarized reflectance (b). The linear distribution of cloudbow oscillations are heterogenous compared to the Figure 4b simulation, and reflect the variability in the cloud field. Both images are RGB composites of pushbrooms from 440, 550, and 670nm view sectors near +38°, presented without axes for visual purposes only. A single gridded pixel in this image represents 50m.*

**Figure 6.** *Several examples of the traditional parametric fit retrieval applied to AirHARP hyper-angular polarized reflectance measurements for 150m superpixels. Plots (a) and (b) signify narrow DSDs, with small CDV values. In plots (c) and (d), the eroded supernumerary bows suggest wider DSDs. Error bars represent the 2-sigma standard deviation of the measurements inside the superpixel. Note that while the $\chi^2red$ in (a) is larger than our 1.5 threshold, overall fit to measurment generates a valid RMSE.*





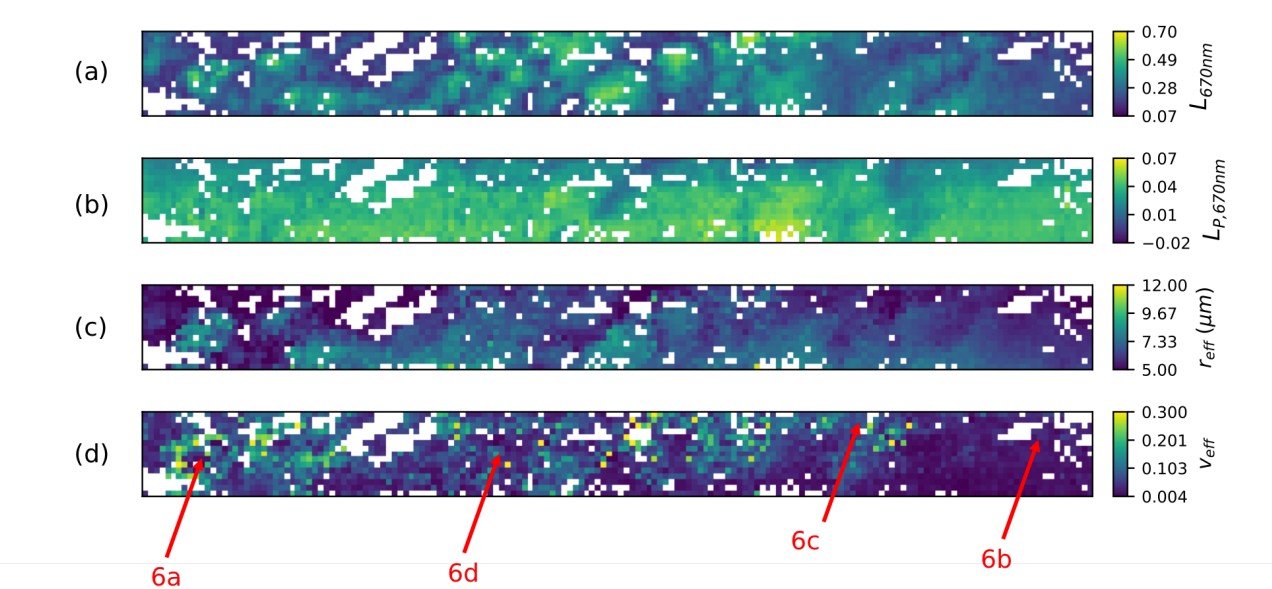

**Figure 7.** *Nadir pushbroom images for 670nm of total intensity (a) and polarized intensity (b), and for the retrieved CDR (c) and CDV(d) for 200m (4x4) gridded superpixels with access to 135-165° in scattering angle, using hyper-angular, co-located data. Quality-flagged retrievals are screened out (white). Note the polarized reflectance is smoother as compared with the reflectance image, and both represent nadir (-0.003°) view sector pushbrooms. The general locations of each retrieval from Figure 6 are identified in red. The scene stretches 3km cross-track and 34km along-track.*

**Figure 8.** *A zoom of two sectors of the AirHARP polarized cloud retrieval for the LMOS cloud field. The intensity image (top) shows both a heterogeneous (a) and a homogenous region (b), defined by the distribution of CDR and CDV, as well as visual cues from the total reflectance. Large-eddy simulations of clean clouds (c), performed by Miller et al. (2018), show that high CDV ($v_{eff}$) and low CDR ($r_{eff}$) typify cloud periphery regions and low CDV and higher CDR occur in the core of the clouds. Similar CDV/CDR relationships are seen in the AirHARP retrievals in (a) at 200m resolution.*



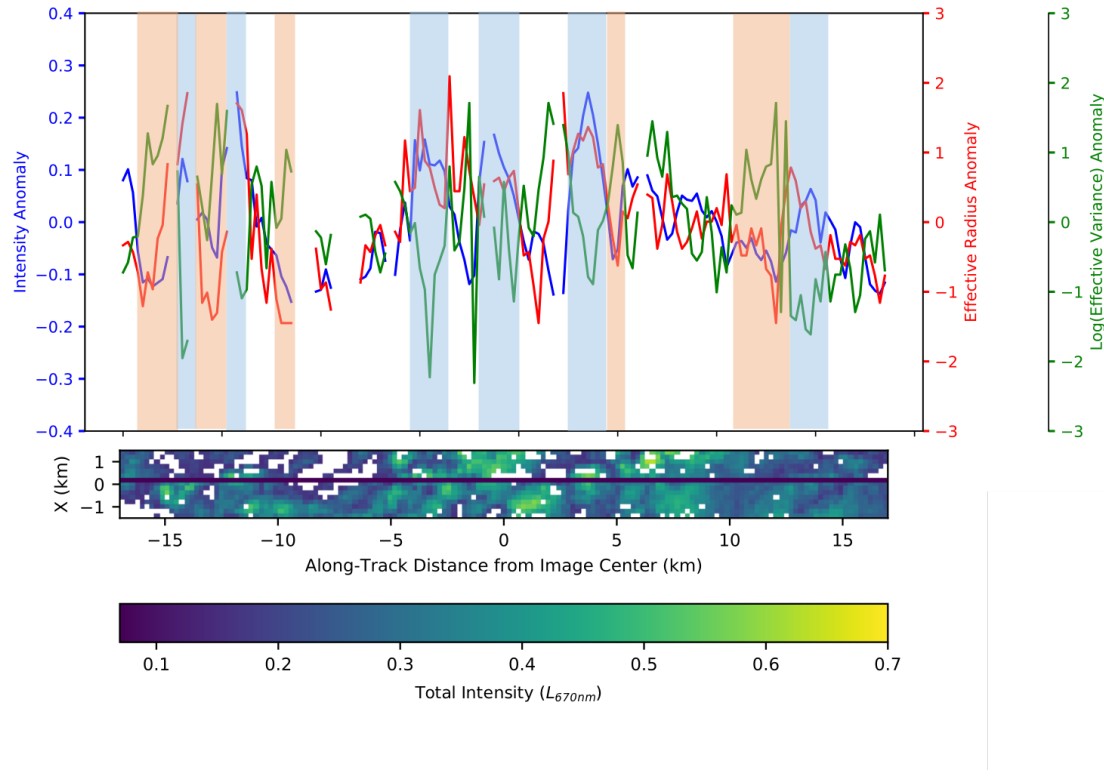

**Figure 9.** *Analysis of intensity (blue), CDR (red), and log(CDV) (green) anomalies from the mean following the black transect along the nadir intensity pushbroom for a segment of the cloud field measured by AirHARP (bottom). Using the spatial distribution of reflectance (intensity) as a proxy for cloud optical thickness (COT), we can visually identify what appears to be cloud cores (blue blocks) and cloud periphery (orange blocks) regions. CDV trends opposite to intensity and CDR in both regions, but wider DSDs with smaller CDR appear at cloud peripheries while larger CDR with narrow DSDs appear in cloud cores. The label X to the left of the intensity image is an abbreviation for cross-track distance from image center line.*









**Figure 10.** *Scale analysis for the scene in Figure 5 with 200m (b) and 600m (c) resolutions. The Mie P12 curves retrieved from the AirHARP data are shown with gray lines on the plots d, e and f, and the direct 600m superpixel P12 retrieval is shown in red for three difference cases: narrow (d), two-regime (e), and mixed (f) DSDs. The boxes to the right-hand side show CDR and CDV results for each of the nine 200m pixel retrievals within the outlined 600m superpixel. Results of the direct retrieval at 600m resolution are shown on the top of the boxes. The location of each retrieval site is given as corresponding colored blocks in the retrieval image for narrow (red), two-regime (peach), and mixed (yellow). Note that the 600m retrievals shown in (e) and (f) give wider CDV results than (d), mainly due to competing size properties at the 200m level.*