# Peer review of "Spatial distribution of cloud droplet size properties from Airborne Hyper-Angular Rainbow Polarimeter (AirHARP) measurements"

_Atmospheric Measurement Techniques, 2019_

## Referee Comment (RC1) · Anonymous Referee #1 · 6 Nov 2019

This paper analyses reflected solar radiance measurement, including its polarized and multi-directional properties, for an estimate of cloud droplet distribution. The retrieval algorithm is applied to airborne measurements. These allows to estimate the cloud droplet radius and cloud droplet variance at unprecedented spatial resolution. The results demonstrate the potential of the measurement for cloud process studies that may be applied to a forthcoming spaceborne instrument.

This paper is of very high quality. It is very clear, present original data, and has potential implications. It can be published with very minor corrections.

I would like the author to add some discussion on the vertical sampling of the technique.

[Figure]

My understanding is that the technique measures up to 3 optical depth, although the majority of the signal comes from the first OD. Is this layer realy representative of the cloud depth ? In particular, I wonder how the modeling results that are shown in Figure 8 are integrated over the vertical for comparison with the radiometric estimate.

Also, I wonder why the authors have limited their analysis to the 670 nm band. The polarized radiance of the cloud shows large spectral variations that are sensitive to the droplet size and variance. Thus, it is unfortunate that the authors have not used this piece of information either to derive additional information, or to make a consistency check based on the variability between the various spectral estimates.

Page 7, line 34 : "in place of Qscat" should be "in place of -Qscat"

---

## Referee Comment (RC2) · Anonymous Referee #2 · 19 Nov 2019

This paper discusses the use of the recently developed polarimeter AirHARP to obtain spatially resolved information about cloud droplet size distributions. First, the instrumental concept and the retrieval approach are described, then a case study is shown, in which AirHARP measurements are used to infer the spatial distribution of cloud effective radius and effective variance over a heterogeneous stratocumulus cloud field observed in 2017. The paper is very well written and shows interesting and novel science, as AirHARP appears capable of producing unique insight on the spatial variability of cloud effective radius and especially effective variance. I fully recommend publication of this paper on AMT, and I think the manuscript really does not need any major changes before publication. Below are a few comments, mostly aimed at improving the

clarity of a few sentences and correcting a few typos.

MINOR COMMENTS

- P1, L13, "uncertainties" -> "sources of uncertainty"

- P2, L7, "depend" -> "depends"

- P2, L10, "are" -> "is"

- P3, L33 (and other instances). I would suggest replacing "confident" with some more specific terms ("precise"? "accurate"?)

- P3, L42. Here, are you referring to retrievals only using one wavelength? What about using multiple wavelengths in order to partially compensate for the lack of angular resolution, exploiting the spectral shift in polarization features?

- P6, L23. Replace "will be launched in 2019" with the actual launch date (it has been recently launched, right?).

- P8, L10. Replace ">" with "<".

- P8, L33-34. "The physics . . . beyond 1.5". Why is it so? Please explain better.

- P8, L37. Add "ratio" after "signal-to-noise"

- P8, L38. Can you briefly explain what optical etaloning is? As a non-instrument person, I don't understand this sentence.

- P9, L30. Add "because" before "the solar geometry"

- P9, L37, "image-to-image" -> "from image to image"

- P10, L2. Do you mean that they tend to miss the angles near the upper end of the cloudbow range? Emphasize this a bit more in the sentence.

- P10, L10. "the actual" -> "RGB composite images of the total and polarized reflectance measured"

[Figure]

- P10, L12. By "the actual image" do you mean the polarized reflectance composite?

- P10, L13-14. "The RGB composite". Isn't the total reflectance image also a RGB composite? If so, I would suggest to say "the polarized reflectance RGB composite" at the beginning of the sentence. Furthermore, does the lower panel of Fig. 5 only shows polarized reflectance, or is also total reflectance superimposed?

- P10, L22. By "standard deviation of the pixels" do you mean the standard deviation of their polarized reflectance?

- P10, L25, "forces a shift to" -> "causes a shift in"

- P13, L12, "the red line" -> "which is shown as a red line"

- P13, L11-13. What about the right hand sides?

- P13, L15. "Mischenko" -> "Mishchenko"

- P13, L27. At least "Fourier" should start in uppercase.

- P13, L30-31. By "well-mixed in CDV" do you mean that they reflect a more heterogeneous distribution of CDV values (I see values ranging from ∼0.015 to ∼0.15)? Say that a bit more clearly.

- P14, L10-11, "angle-to-angle" -> "from angle to angle"

---

## Referee Comment (RC3) · Gerard van Harten (Referee) · 27 Nov 2019

RECOMMENDATION:

Minor revisions

GENERAL COMMENTS:

The authors present the retrieval of cloud droplet size distributions from Airborne Hyper-Angular Rainbow Polarimeter (AirHARP) data collected during the Lake Michigan Ozone Study (LMOS) field campaign. The scientific relevance of the work is clearly explained, as well as the timeliness with HARP Cubesat launched earlier this month

and HARP2 in development for the 2022 PACE mission. The radiometric bi-spectral retrieval, and the multi-angle polarimetric retrieval methods are introduced, followed by an overview of multi-angle polarimetric instruments and the strengths and weaknesses of their cloud retrievals, based on how each instrument uniquely samples the parameter space. The AirHARP instrument is described, particularly focussing on the features relevant for cloud droplet size retrievals. Next, the retrieval framework is presented, including the fit quality metrics. The large dimensionality of the AirHARP data is carefully explained using simulated imagery and LMOS data. Despite the limited size of the LMOS dataset over clouds, a homogeneous and a heterogeneous region are identified based on intensity (L) and droplet size effective radius (CDR) and variance (CDV). Cloud cores are associated with high L, high CDR, and low CDV, whereas cloud periphery regions are associated with the opposite. These findings are consistent with large-eddy simulations by Miller et al. (2018). An analysis follows of the impact of instrument spatial resolution on droplet size retrievals by aggregating the polarized reflectances into 600x600 instead of 200x200 m2 superpixels, showing broadening of CDV in a heterogeneous region. Finally, limitations and error sources are discussed, including pixel-level coregistration and optical etaloning in the imagery. Several recommendations for future work are provided, including field campaigns dedicated to clouds; comparisons with HARP Cubesat, HARP2, GOES-R; etaloning correction using science data; etaloning correction using HARP2 onboard calibrator; multi-modal retrieval algorithm; multi-wavelength retrieval algorithm; pixel-level topographic algorithm. The paper is very well written, with great attention to detail in text and graphics.

SPECIFIC COMMENTS:

AirHARP's native pixel size is 50x50 m2. However, data and retrievals presented in the paper are from 200x200 m2 superpixels.

I encourage the authors to reconsider showing retrievals at 50m resolution, given the specific attention in the paper to the impact of spatial resolution on droplet size retrievals, and statements such as:

[Figure]

"Our retrievals from this dataset show that cloud DSD heterogeneity can occur at the 200m scale, much smaller than the 1-2km resolution of most spaceborne sensors. This heterogeneity at the subpixel level can create artificial broadening of the DSD in retrievals made at resolutions on the order of 0.5 to 1 km." (P1_31)

The motivation for binning to 200m is "to increase SNR and mitigate other potential artifacts in the data. These artifacts will be discussed in Section 6" (P10_21).

However, it is not clear to me from Section 6 what exactly is the problem at 50m resolution, and the positive results at 200m rise the question if 50m could still be usable for the paper:

Section 6 (P14_21): "Conservative cloud identification and binning pixels to 200m (4x4) resolution further mitigates the error introduced by using this mean height. ... all retrievals shown successfully fit the RMSE threshold defined above. Therefore, we believe errors in our geolocation do not contribute significantly to the results of our study."

DETAILED COMMENTS:

- P2_15: "clouds would not exist at the scale we see them today": vague

- P3_21: "Reidi" -> "Riedi"

- P4_22: "fourier" -> "Fourier"

- P4_35: "modeled correlations" -> "image-specific empirical correlations"

- P9_11: "These diagnostics ... of the LUT": hard to read

- P11_24: "... retrieved on the periphery": Point out that reduced cloud fraction should not impact polarimetric retrieval (see Eg. (4))

- P12_38: "consistent with current research and theories of cloud microphysics": Similar finding with AirMSPI (Fig. 13 in Xu et al. (2018))

- P13_24: "fourier" -> "Fourier"

- Figure 5: Indicate flight direction, because it is perpendicular to Fig. 4b

- Figure 5, caption: What is the image size?

- Figure 6, caption: "150m superpixels", whereas: P10_18: "Figure 6 shows several examples of an AirHARP 200m superpixel retrieval" Fig. 6: text inside plots "200m grd.res."

- Figure 9: Units missing for Intensity and Effective Radius

---

## Author Comment (AC2) · 14 Jan 2020

AR #1:

"Page 7, line 34 : "in place of Qscat" should be "in place of -Qscat"

Author response:

The requested change will be made as recommended.

---

## Author Comment (AC3) · 14 Jan 2020

The authors thank Anonymous Reviewer #2 (AR #2) for their comments and review of the manuscript. The following author responses are given below the AR #2 comments for ease.

— AR #2:—

- P1, L13, "uncertainties" -> "sources of uncertainty"

- P2, L7, "depend" -> "depends"

- P2, L10, "are" -> "is"

[Figure]

- P3, L33 (and other instances). I would suggest replacing "confident" with some more specific terms ("precise"? "accurate"?)

Author Response:

These changes will be made as noted in the manuscript. All instances of "confident" will be changed to "accurate".

— AR #2:—

- P3, L42. Here, are you referring to retrievals only using one wavelength? What about using multiple wavelengths in order to partially compensate for the lack of angular resolution, exploiting the spectral shift in polarization features?

Author Response:

This line is talking about polarized retrievals done using a single wavelength, though one can use multi-spectral sampling to do as AR #2 suggests. The benefit of multi-spectral sampling is touched on briefly later in the same section. There is evidence in the literature (Alexandrov papers, Miller et al. 2018, Shang et al. 2015, our study, etc.) that a single wavelength is enough to retrieve CDR and CDV properties, as long as that channel samples the cloudbow with sufficient angular coverage.

We will make this more explicit by adding to the opening sentence (P3, L32, * is the change): " Multiangle sampling at high angular density and moderate pixel resolution are essential elements of a accurate, *single-wavelength* retrieval."

— AR #2:—

- P6, L23. Replace "will be launched in 2019" with the actual launch date (it has been recently launched, right?).

- P8, L10. Replace ">" with "<".

Author Response:

The changes will be made as recommended. The HARP CubeSat launched on Nov. 2 2019. This will be updated.

— AR #2:—

- P8, L33-34. "The physics . . . beyond 1.5". Why is it so? Please explain better.

Author Response:

We will add the following sentences in to P9,L10 before "Third, Breon and Goloub...":

"Because the x2red depends strongly on the uncertainty of the individual measurements, there is also a possibility that pixels that represent narrow size distributions may give a valid retrieval, while producing x2red values beyond 1.5. Figure 6a is one such example. The cloudbow oscillations are well-defined and AirHARP data clearly captures the pattern, though the x2red is 2.52. While the error bar on several AirHARP data points does not touch the best fit polarized reflectance, the overall curve fit does represent the information content in the measurement. It is therefore important to include the RMSE as a two-factor authentication. The RMSE evaluates how close the data points are to the best fit curve, with no regard to measurement uncertainty.

— AR #2:—

- P8, L37. Add "ratio" after "signal-to-noise"

- P8, L38. Can you briefly explain what optical etaloning is? As a non-instrument person, I don't understand this sentence.

- P9, L30. Add "because" before "the solar geometry"

- P9, L37, "image-to-image" -> "from image to image"

Author Response:

Minor changes will be made as noted. Instead of adding an explanation on optical etaloning, much of which is outside the scope of the paper, the authors will add a

technical citation here for those seeking more information:

andor.oxinst.com: Oxford Instruments [online] Available from: https://andor.oxinst.com/learning/view/article/optical-etaloning-in-charge-coupled-devices (Accessed 14 Jan 2020), ______.

And will reference in-text as: (andor.oxinst.com,____)

— AR #2:—

- P10, L2. Do you mean that they tend to miss the angles near the upper end of the cloudbow range? Emphasize this a bit more in the sentence.

Author response:

Yes, but perhaps it is not clear enough. We will change "Targets observed outside these lines do not access the full cloudbow scattering angle range (<165°)." to "This work does not perform a retrieval on any targets observed outside these lines. Outside these lines, the reduced scattering angle coverage at the upper end of the cloudbow range begins to truncate the signal from the supernumerary bows. Because the majority of the size distribution information is encoded in the supernumerary bows (145-165 deg), it is important that the full scattering angle range is preserved."

— AR #2:—

- P10, L10. "the actual" -> "RGB composite images of the total and polarized reflectance measured"

- P10, L12. By "the actual image" do you mean the polarized reflectance composite?

- P10, L13-14. "The RGB composite". Isn't the total reflectance image also a RGB composite? If so, I would suggest to say "the polarized reflectance RGB composite" at the beginning of the sentence. Furthermore, does the lower panel of Fig. 5 only shows polarized reflectance, or is also total reflectance superimposed?

- P10, L22. By "standard deviation of the pixels" do you mean the standard deviation of their polarized reflectance?

- P10, L25, "forces a shift to" -> "causes a shift in"

- P13, L12, "the red line" -> "which is shown as a red line"

Author Response:

The authors will change "the actual image" and "RGB composite" in this section to "polarized reflectance image" for clarity. They are both RGB composites, as AR #2 notes. We apologize for any confusion here. Both images in Figure 5 are RGB composites, and total reflectance image is not superimposed on the polarized reflectance image.

And yes, AR #2 is correct in the interpretation of P10,L22. This sentence will be changed from "standard deviation of the pixels" to "standard deviation of the polarized reflectance measured at the pixels" for clarity.

All other minor corrections will be made as noted.

— AR #2:—

- P13, L11-13. What about the right hand sides?

Author Response:

Excellent catch, we recognize we did not explicitly explain those parts of the figure. The authors will add this segment after P13, L13 "...retrieval, the red line.":

"The two boxes to the right of each of the retrieved P12 curve plots in Figure 10d-f represent the retrieved CDR (middle column) and CDV (right column) for the colored superpixel boxes located in Figure 10a-c. The 600m CDR or CDV result is given in the title above each box and represents the retrieval for the entire 9-box square underneath, whereas the 200m CDR or CDV results are shown inside each colored sub-box." We will also move the P13,L13 "Figure 10d shows that..." into the next paragraph.

— AR #2:—

- P13, L15. "Mischenko" -> "Mishchenko"

- P13, L27. At least "Fourier" should start in uppercase.

- P13, L30-31. By "well-mixed in CDV" do you mean that they reflect a more heterogeneous distribution of CDV values (I see values ranging from âĹij0.015 to âĹij0.15)? Say that a bit more clearly.

- P14, L10-11, "angle-to-angle" -> "from angle to angle"

Author Response:

All minor comments will be changed as noted. By "well-mixed in CDV", the authors mean that there is a larger distribution of CDV values as compared to Figures 10d-e, not necessarily a heterogenous distribution. There are two meanings of heterogenous: the CDV value itself is a measure of droplet size heterogeneity in the pixel and a distribution of CDV values can be heterogenous if there are one or two clusters of values inside the same superpixel bin. It is important for the paper to be consistent to avoid confusion. When the word "heterogenous" is used in this work, it always refers to a high single CDV value in the retrieval (indicative of many droplet sizes existing inside the same pixel) or a visual variability in a cloud field. Here, the authors will rephrase this comment from "well-mixed in CDV" to "the retrieved 200m P12 curves show a wider spread of CDV values, as compared to the results shown in Figures 10d-e."

---

## Author Response (AR1)

**Reply to Anonymous Reviewer #1**

The authors thank Anonymous Reviewer #1 (AR #1) for their thorough review of the manuscript. The reviewer's comment will be outlined in quotes, bolded, and italicized and the author response will be given underneath in plain text.

***"My understanding is that the technique measures up to 3 optical depth, although the majority of the signal comes from the first OD. Is this layer really representative of the cloud depth ? In particular, I wonder how the modeling results that are shown in Figure 8 are integrated over the vertical for comparison with the radiometric estimate."***

This layer is not representative of the total cloud depth (except for clouds on the order of COD ~3), but the sizes retrieved there serve as a tracer for microphysical processes going on toward the cloud top (i.e. condensation) and potential complexity on the cloud periphery. As per AR #1's recommendation, we will be more explicit in the paper about the penetration depth of the polarized signal and how representative it is for the entire cloud depth.

As per AR #1's second point, the "ATEX clean" simulation (Figure 8c) is described in detail in Miller et al. (2018), already takes into account the vertical weighting of the droplet size distribution in each grid box. Miller et al. (2018) do a sensitivity study, shown in Figure 3 of that paper, that compares a radiometric retrieval of CDR and COT on the LES field to the vertically-weighted parameters of the LES simulation and the results are quite good. When a theoretical polarimetric retrieval is applied to the same LES simulation, the correlations between retrieved and vertically-weighted CDR and CDV are again, quite good (Figure 4 in Miller et al. (2018)). This result suggests that vertical weighting techniques can theoretically reproduce both the polarimetric, single scattering signal at cloud top and the radiometric signal, which comes from multiple scattering throughout the cloud.

We would ideally need a co-incident radiometer with comparable resolution and necessary channels for a data-based study between radiometer and AirHARP estimates of droplet size properties, as discussed in the manuscript. This scenario was not available for this dataset, unfortunately. We anticipate future AirHARP aircraft campaigns with a cloud focus such that we can provide a data-driven study of this kind.

***"Also, I wonder why the authors have limited their analysis to the 670 nm band. The polarized radiance of the cloud shows large spectral variations that are sensitive to the droplet size and variance. Thus, it is unfortunate that the authors have not used this piece of information either to derive additional information, or to make a consistency check based on the variability between the various spectral estimates."***

The authors did not limit our study to the 670nm band, as AR #1 suggests. In fact, it was the opposite; the HARP instrument was proposed to perform hyper-angular cloud retrievals at up to 60 viewing angles at 670nm, at the pixel-level and across a spatial field. This paper serves as the validation of this new technology. Furthermore, our study shows that a single polarized wavelength, with sufficient viewing angles, is enough to retrieve cloud droplet size properties. There is supporting evidence in the literature (Alexandrov et al. 2016 and other referred papers from this author) and Mie scattering curves at different wavelengths are degenerate for a given CDR and CDV (Figure 1a).

AR #1 suggests that using multiple channels could serve as a consistency check, however, it is possible that the lack of along-track angular coverage in the other three AirHARP channels makes the retrieval less robust than the 670nm hyper-angular method presented in this paper. The other three AirHARP bands are not hyper-angular: at 20 distinct view angles each, they sample the cloudbow at the same pixel three times less frequently than at 670nm. The limited coverage in these other channels may not always capture the cloudbow oscillations, especially for narrow size distributions. Therefore, the non-670nm channels may introduce more uncertainty to a joint retrieval than what already exists in the hyper-angular 670nm retrieval. On the other hand, adding the other channels may provide more information about above-cloud aerosol, Rayleigh scattering, or improve thermodynamic phase or multi-layer cloud discrimination. This complexity requires its own sensitivity study and would distract from the focus of this paper, if included.

As discussed in the manuscript, the joint spectral retrieval can also be done in a completely different way: cross-track, instead of along-track, similar to the method in Xu et al. (2018). This concept alone warrants its own study: while the underlying retrieval method is the same, the way it is done is wholly orthogonal and contains uncertainties different to the method presented in this paper.

*"Page 7, line 34 : "in place of Qscat" should be "in place of -Qscat"*

The requested change will be made as recommended.

**Reply to Anonymous Referee #2**

The authors thank Anonymous Reviewer #2 (AR #2) for their thorough review of the manuscript. The reviewer's comment will be outlined in quotes, bolded, and italicized and the author response will be given underneath in plain text.

*"- P1, L13, "uncertainties" -> "sources of uncertainty"*
 *- P2, L7, "depend" -> "depends"*
 *- P2, L10, "are" -> "is"*
 *- P3, L33 (and other instances). I would suggest replacing "confident" with some more specific terms ("precise"? "accurate"?)"*

These changes will be made as noted in the manuscript. All instances of "confident" will be changed to "accurate".

*"- P3, L42. Here, are you referring to retrievals only using one wavelength? What about using multiple wavelengths in order to partially compensate for the lack of angular resolution, exploiting the spectral shift in polarization features?"*

This line is talking about polarized retrievals done using a single wavelength, though one can use multi-spectral sampling to do as AR #2 suggests. The benefit of multispectral sampling is touched on briefly later in the same section. There is evidence in the literature (Alexandrov papers, Miller et al. 2018, Shang et al. 2015, our study, etc.) that a single wavelength is enough to retrieve CDR and CDV properties, as long as that channel samples the cloudbow with sufficient angular coverage. We will make this more explicit by adding to the opening sentence (P3, L32, * is the change): " Multiangle sampling at high angular density and moderate pixel resolution are essential elements of a accurate, *single-wavelength* retrieval."

*" - P6, L23. Replace "will be launched in 2019" with the actual launch date (it has been recently launched, right?).*
 *- P8, L10. Replace ">" with "<". "*

The changes will be made as recommended. The HARP CubeSat launched on Nov. 2 2019. This will be updated.

*" - P8, L33-34. "The physics . . . beyond 1.5". Why is it so? Please explain better."*

We will add the following sentences in to P9,L10 before "Third, Breon and Goloub...": "Because the x2red depends strongly on the uncertainty of the individual measurements, there is also a possibility that pixels that represent narrow size distributions may give a valid retrieval, while producing x2red values beyond 1.5. Figure 6a is one such example. The cloudbow oscillations are well-defined and AirHARP

data clearly captures the pattern, though the x2red is 2.52. While the error bar on several AirHARP data points does not touch the best fit polarized reflectance, the overall curve fit does represent the information content in the measurement. It is therefore important to include the RMSE as a two-factor authentication. The RMSE evaluates how close the data points are to the best fit curve, with no regard to measurement
5  uncertainty."

*"- P8, L37. Add "ratio" after "signal-to-noise"*
* - P8, L38. Can you briefly explain what optical etaloning is? As a non-instrument person, I don't understand this sentence.*
10  *- P9, L30. Add "because" before "the solar geometry"*
* - P9, L37, "image-to-image" -> "from image to image""*

Minor changes will be made as noted. Instead of adding an explanation on optical etaloning, much of which is outside the scope of the paper, the authors will add a C3 technical citation here for those seeking
15  more information:

andor.oxinst.com:       Oxford       Instruments     [online]     Available     from: https://andor.oxinst.com/learning/view/article/optical-etaloning-in-charge-coupleddevices (Accessed 14 Jan 2020), _____.
20
And will reference in-text as: (andor.oxinst.com,____)

*" - P10, L2. Do you mean that they tend to miss the angles near the upper end of the cloudbow range? Emphasize this a bit more in the sentence."*

Yes, but perhaps it is not clear enough. We will change "Targets observed outside these lines do not access the full cloudbow scattering angle range (<165∘)." to "This work does not perform a retrieval on any targets observed outside these lines. Outside these lines, the reduced scattering angle coverage at the upper end of the cloudbow range begins to truncate the signal from the supernumerary bows. Because the
30  majority of the size distribution information is encoded in the supernumerary bows (145-165 deg), it is important that the full scattering angle range is preserved."

*"- P10, L10. "the actual" -> "RGB composite images of the total and polarized reflectance measured"*
* - P10, L12. By "the actual image" do you mean the polarized reflectance composite?*
35  *- P10, L13-14. "The RGB composite". Isn't the total reflectance image also a RGB*
*composite? If so, I would suggest to say "the polarized reflectance RGB composite" at*
*the beginning of the sentence. Furthermore, does the lower panel of Fig. 5 only shows*
*polarized reflectance, or is also total reflectance superimposed?*
* - P10, L22. By "standard deviation of the pixels" do you mean the standard deviation of their polarized*
40  *reflectance?*
* - P10, L25, "forces a shift to" -> "causes a shift in"*
* - P13, L12, "the red line" -> "which is shown as a red line""*

The authors will change "the actual image" and "RGB composite" in this section to "polarized reflectance image" for clarity. They are both RGB composites, as AR #2 notes. We apologize for any confusion here. Both images in Figure 5 are RGB composites, and total reflectance image is not superimposed on the polarized reflectance image. And yes, AR #2 is correct in the interpretation of P10, L22. This sentence will be changed from "standard deviation of the pixels" to "standard deviation of the polarized reflectance measured at the pixels" for clarity. All other minor corrections will be made as noted.

*" - P13, L11-13. What about the right hand sides?"*

Excellent catch, we recognize we did not explicitly explain those parts of the figure. The authors will add this segment after P13, L13 "...retrieval, the red line.": "The two boxes to the right of each of the retrieved P12 curve plots in Figure 10d-f represent the retrieved CDR (middle column) and CDV (right column) for the colored superpixel boxes located in Figure 10a-c. The 600m CDR or CDV result is given in the title above each box and represents the retrieval for the entire 9-box square underneath, whereas the 200m CDR or CDV results are shown inside each colored sub-box." We will also move the P13,L13 "Figure 10d shows that..." into the next paragraph.

*" - P13, L15. "Mischenko" -> "Mishchenko"*
*- P13, L27. At least "Fourier" should start in uppercase.*
*- P13, L30-31. By "well-mixed in CDV" do you mean that they reflect a more heterogeneous distribution of CDV values (I see values ranging from âĹij0.015 to âĹij0.15)? Say that a bit more clearly.*
*- P14, L10-11, "angle-to-angle" -> "from angle to angle"*

All minor comments will be changed as noted. By "well-mixed in CDV", the authors mean that there is a larger distribution of CDV values as compared to Figures 10d-e, not necessarily a heterogenous distribution. There are two meanings of heterogenous: the CDV value itself is a measure of droplet size heterogeneity in the pixel and a distribution of CDV values can be heterogenous if there are one or two clusters of values inside the same superpixel bin. It is important for the paper to be consistent to avoid confusion. When the word "heterogenous" is used in this work, it always refers to a high single CDV value in the retrieval (indicative of many droplet sizes existing inside the same pixel) or a visual variability in a cloud field. Here, the authors will rephrase this comment from "well-mixed in CDV" to "the retrieved 200m P12 curves show a wider spread of CDV values, as compared to the results shown in Figures 10d-e."

**Reply to Gerard van Harten (Reviewer #3)**

The authors thank Gerard van Harten (Reviewer #3) for his thorough review of the manuscript. The reviewer's comment will be outlined in quotes, bolded, and italicized and the author response will be given underneath in plain text.

*"AirHARP's native pixel size is 50x50 m2. However, data and retrievals presented in the paper are from 200x200 m2 superpixels. I encourage the authors to reconsider C1 showing retrievals at 50m resolution, given the specific attention in the paper to the impact of spatial resolution on droplet size retrievals, and statements such as:*

*"Our retrievals from this dataset show that cloud DSD heterogeneity can occur at the 200m scale, much smaller than the 1-2km resolution of most spaceborne sensors. This heterogeneity at the subpixel level can create artificial broadening of the DSD in retrievals made at resolutions on the order of 0.5 to 1 km." (P1_31)*

*The motivation for binning to 200m is "to increase SNR and mitigate other potential artifacts in the data. These artifacts will be discussed in Section 6" (P10_21). However, it is not clear to me from Section 6 what exactly is the problem at 50m resolution, and the positive results at 200m rise the question if 50m could still be usable for the paper:*

*Section 6 (P14_21): "Conservative cloud identification and binning pixels to 200m (4x4) resolution further mitigates the error introduced by using this mean height. . . . all retrievals shown successfully fit the RMSE threshold defined above. Therefore, we believe errors in our geolocation do not contribute significantly to the results of our study."*

The authors were excited to see GvH's encouraging recommendation and perspective here. We will explain why this idea was initially considered for the paper but ultimately not done.

First, this paper wears several hats. It is the debut of the HARP instrument concept in the peer-reviewed literature, and so must (1) introduce the instrument, (2) provide the rationale for existing alone and alongside other modern cloud-measuring instruments, (3) make a firm case that these cloud measurements will and/or will not translate to space-borne platforms, and (4) ensure quality-assurance of the results, and (5) create a focused and concise story. These five major points, especially (3) and (4), are what contributed to cutting this study from the paper.

It is important to note that we can perform the cloud retrieval at 50m on this AirHARP dataset. In fact, we did during the testing and evaluation of the retrieval, and we encourage the community to explore AirHARP data at this resolution. There are several factors that led us to using 200m retrievals in this story:

(1) Is it useful to show 50m retrievals in a paper that uses AirHARP measurements as a proxy for HARP CubeSat and HARP2? These two instruments will not achieve 50m ground resolution from orbit; they are expected to see 400 and 700m native pixel resolution at nadir (and will continue bin for SNR and data storage). Therefore, we chose to show the retrievals at 200m: we can highlight the unprecedented, co-located resolution of HARP for this kind of retrieval and stay comparable to the pixel sizes of these other space-borne sensors with minimal binning. We can also extend this comparison to radiometric data. Our retrieval resolution is on the same order as today's operational L2 cloud products (i.e. MODIS, VIIRS, ABI).

We also felt that adding this study either together with the 200m retrievals or alone would put a larger focus on AirHARP, rather than framing the AirHARP results shown here as a proxy for HARP CubeSat and HARP2. The authors are concerned that this would detract from the overall message of the paper. We will change the paper text to make this message more obvious:

P1, L34: "This AirHARP study demonstrates 35 the viability of the HARP concept to make cloud measurements at scales of individual clouds with global coverage, and all in a low-cost, compact CubeSat-size payload." was changed to "This study, which uses the AirHARP instrument and its data as a proxy for upcoming HARP CubeSat and HARP2 spaceborne instruments, demonstrates the viability of the HARP concept to make cloud measurements at scales of individual clouds, with global coverage, and in a low-cost, compact CubeSat-size payload."

P5, L6: "In this paper, we will first describe the HARP concept, with a focus on the AirHARP instrument and its data as a proxy for upcoming HARP CubeSat and HARP2 C3 space instruments." was changed to "In this paper, we will first describe the HARP concept, and frame the AirHARP instrument and its data as a proxy for upcoming HARP CubeSat and HARP2 space instruments throughout the rest of the work."

(2) We estimate our error in geolocating the cloud deck altitude at +-50m in the vertical, as mentioned in Section 6. Doing retrievals at the same scale may not maintain the quality of those done at larger superpixel sizes. In this direction, the larger 200m (and even 600m) superpixel size increases measurement SNR, mitigates geolocation errors, and allows us to use superpixel standard deviation to evaluate our retrieval fits (x2red). We will change the paper text to make this more obvious:

P14, L31: "Therefore, we believe errors in our geolocation do not contribute significantly to the results of our study." was changed to "Therefore, we believe errors in our geolocation do not contribute significantly to the results of our study shown at 200m or 600m resolutions."

The authors ultimately decided to keep the paper concise and focused, without sacrificing information content. There are many opportunities to expand this work in the future, especially with new AirHARP campaigns that specifically target clouds with other comparable instruments and as our data correction algorithms mature. We again thank GvH for his recommendations.

*" - P2_15: "clouds would not exist at the scale we see them today": vague*

*- P3_21: "Reidi" -> "Riedi"*
*- P4_22: "fourier" -> "Fourier"*
*- P4_35: "modeled correlations" -> "image-specific empirical correlations"*

5    All minor corrections will be changed as noted. P2,L15: "These particles seed both liquid water and ice clouds in our atmosphere in a process that is so important to cloud development that without aerosols, clouds would not exist at the scale we see them today." was removed. Instead, P2, L12: "Aerosols drop the energy barrier required for condensation, serving as cloud condensation nuclei (Petters and Kreidenweis 2007)." was changed to "Aerosols drop the energy barrier required for condensation, serving

10    as condensation nuclei for liquid water and ice clouds in our atmosphere (Petters and Kreidenweis 2007)."

*" - P9_11: "These diagnostics . . . of the LUT": hard to read"*

P9, L14: "These diagnostics also account for artifacts from rotating into the scattering plane 15 on Qsca

15    and Usca and retrievals that poorly converge to CDR and CDV values at the very edge of the LUT." was changed to "These diagnostics also account for any artifacts that arise from rotating our reference frame of polarization into the scattering plane. Retrievals that converge artificially to the edges of the LUT are also screened by the RMSE and x2red."

20    *" - P11_24: ". . . retrieved on the periphery": Point out that reduced cloud fraction should not impact polarimetric retrieval (see Eg. (4)) "*

This is noted explicitly when describing the retrieval process itself on P8, L13. We will change P8, L11 from "Corrective factors for aerosol above cloud, cirrus, sun glint, molecular scattering, and surface

25    reflectance signals comprise weak functions of scattering angle, with the parameter [alpha] related to cloud fraction (Breon and Goloub 1998, Diner et al. 2013, Alexandrov et al. 2015)." to "Corrective factors for aerosol above cloud, cirrus, sun glint, molecular scattering, and surface reflectance signals C5 comprise weak functions of scattering angle (Diner et al. 2013, Alexandrov et al. 2015). The parameter [alpha] is related to cloud fraction (Breon and Goloub 1998), and therefore, is also accounted for by Eq.

30    (4)."

*" - P12_38: "consistent with current research and theories of cloud microphysics": Similar finding with AirMSPI (Fig. 13 in Xu et al. (2018))*
*- P13_24: "fourier" -> "Fourier"*

Minor corrections are changed as noted. As this sentence (P12, L38) is a summary sentence that is meant to close the paragraph and reference the research mentioned in the paragraph above, we will not add any references at the end. Xu et al. (2018) is heavily sourced in the text in other sections.

40    *"- Figure 5: Indicate flight direction, because it is perpendicular to Fig. 4b*
*- Figure 5, caption: What is the image size?*

*- Figure 6, caption: "150m superpixels", whereas: P10_18: "Figure 6 shows several examples of an AirHARP 200m superpixel retrieval" Fig. 6: text inside plots "200m grd.res."*
*- Figure 9: Units missing for Intensity and Effective Radius"*

5     All recommendations are valid and changes will be made as noted. The image size of Figure 5 is approximately 37km by 5km and will be added to the figure caption.

**Ordered List of Changes**

- P1, L13: "uncertainties" changed to "sources of uncertainty"

- P1, L34: "This AirHARP study demonstrates the viability of the HARP concept to make cloud measurements at scales of individual clouds with global coverage, and all in a low-cost, compact CubeSat-size payload." changed to "This study, which uses the AirHARP instrument and its data as a proxy for upcoming HARP CubeSat and HARP2 spaceborne instruments, demonstrates the viability of the HARP concept to make cloud measurements at scales of individual clouds, with global coverage, and in a low-cost, compact CubeSat-size payload."

- P2, L7: "depend" changed to "depends"

- P2, L10: "are" changed to "is"

- P2, L12: "Aerosols drop the energy barrier required for condensation, serving as cloud condensation nuclei (Petters and Kreidenweis 2007)." was changed to "Aerosols drop the energy barrier required for condensation, serving as condensation nuclei for liquid water and ice clouds in our atmosphere (Petters and Kreidenweis 2007)."

- P2,L15: "These particles seed both liquid water and ice clouds in our atmosphere in a process that is so important to cloud development that without aerosols, clouds would not exist at the scale we see them today." was removed.

- P3,L21: "Reidi" changed to "Riedi"

- P3, L32: " Multiangle sampling at high angular density and moderate pixel resolution are essential elements of a accurate, retrieval." changed to " Multiangle sampling at high angular density and moderate pixel resolution are essential elements of an accurate, single wavelength retrieval."

- P3, L33 and P4, L8: "confident" changed to "accurate"

- P4,L22: "fourier" changed to "Fourier"

- P4,L35: "modeled correlations" changed to "image-specific empirical correlations"

- P5, L6: "In this paper, we will first describe the HARP concept, with a focus on the AirHARP instrument and its data as a proxy for upcoming HARP CubeSat and HARP2 C3 space

instruments." was changed to "In this paper, we will first describe the HARP concept, and frame the AirHARP instrument and its data as a proxy for upcoming HARP CubeSat and HARP2 space instruments throughout the rest of the work."

- P6, L23: "will be launched in 2019" changed to "was launched on November 2, 2019"
- P6, L23: removed "orbit"
- P6, L23: "inclination" changed to "inclined orbit"
- P6, L24: added "in February 2020"
- P6,L25: "HARP CubeSat will perform cloud retrievals" changed to "Cloud retrievals on HARP CubeSat data will be possible" (phrasing error missed by all reviewers)
- P7, L34: "in place of Qscat" changed to "in place of -Qscat"
- P8, L10: ">" changed to "<"
- P8, L11: "Corrective factors for aerosol above cloud, cirrus, sun glint, molecular scattering, and surface reflectance signals comprise weak functions of scattering angle, with the parameter [alpha] related to cloud fraction (Breon and Goloub 1998, Diner et al. 2013, Alexandrov et al. 2015)." changed to "Corrective factors for aerosol above cloud, cirrus, sun glint, molecular scattering, and surface reflectance signals comprise weak functions of scattering angle (Diner et al. 2013, Alexandrov et al. 2015). The parameter [alpha] is related to cloud fraction (Breon and Goloub 1998), and therefore, is accounted for by Eq. (4)."
- P8, L37: added "ratio" after "signal-to-noise"
- P8, L38: added new reference in text (andor.oxinst.com,____) with citation added in References: andor.oxinst.com: Oxford Instruments [online] Available from: https://andor.oxinst.com/learning/view/article/optical-etaloning-in-charge-coupleddevices (Accessed 14 Jan 2020), _____.
- P9, L10: before "Third, Breon and Goloub..." added "Because the x2red depends strongly on the uncertainty of the individual measurements, there is also a possibility that pixels that represent narrow size distributions may give a valid retrieval, while producing x2red values beyond 1.5. Figure 6a is one such example. The cloudbow oscillations are well-defined and AirHARP data clearly captures the pattern, though the x2red is 2.52. While the error bar on several AirHARP

data points does not touch the best fit polarized reflectance, the overall curve fit does represent the information content in the measurement. It is therefore important to include the RMSE as a two-factor authentication." The last sentence in the original author response was removed for brevity.

- P9, L14: "These diagnostics also account for artifacts from rotating into the scattering plane 15 on Qsca and Usca and retrievals that poorly converge to CDR and CDV values at the very edge of the LUT." was changed to "These diagnostics also account for any artifacts that arise from rotating our reference frame of polarization into the scattering plane and retrievals that converge artificially to the edges of the LUT." The wording here was changed slightly from the author response.

- P9, L30, added "because" before "the solar geometry" The authors disagree with the grammatical suggestion here, different from the original author response. No changes were made.

- P9, L37: "image-to-image" changed to "from image to image""

- P10, L2: "Targets observed outside these lines do not access the full cloudbow scattering angle range (<165∘)." changed to "This work does not perform a retrieval on any targets observed outside these lines. Outside these lines, the reduced scattering angle coverage at the upper end of the cloudbow range begins to truncate the signal from the supernumerary bows. Because the majority of the size distribution information is encoded in the supernumerary bows (145-165 deg), it is important that the full scattering angle range is preserved."

- P10,L9: added "during LMOS, in total (top) and polarized reflectance (bottom),"

- P10,L10: removed ", during the LMOS campaign."

- P10, L10: "the actual" changed to "RGB composite images of the total and polarized reflectance measured"

- P10, L12: "the actual image" changed to "polarized reflectance image"

- P10, L13-14: "the RGB composite" changed to "polarized reflectance image"

- P10, L22: By "standard deviation of the pixels" changed to "standard deviation of the polarized reflectance measured at the pixels"

- P10, L25: "forces a shift to" changed to "causes a shift in"

- P10, L40: "um-1" changed to "nm$^{-1}$" (minor error missed by all reviewers)
- P13, L12: "the red line" changed to "which is shown as a red line""
- P13, L13: after "...retrieval, the red line." added "The two boxes to the right of each of the retrieved P12 curve plots in Figure 10d-f represent the retrieved CDR (middle column) and CDV (right column) for the colored superpixel boxes located in Figure 10a-c. The 600m CDR or CDV result is given in the title above each box and represents the retrieval for the entire 9-box square underneath, whereas the 200m CDR or CDV results are shown inside each colored sub-box." P13,L13 "Figure 10d shows that..." was moved into the next paragraph.
- P13, L15: "Mischenko" changed to "Mishchenko"
- P13, L27: "fourier" changed to "Fourier"
- P13, L30-31: "well-mixed in CDV" changed to "the retrieved 200m P12 curves show a wider spread of CDV values, as compared to the results shown in Figures 10d-e."
- P14, L10-11: "angle-to-angle" changed to "from angle to angle"
- P14, L31: "Therefore, we believe errors in our geolocation do not contribute significantly to the results of our study." was changed to "Therefore, we believe errors in our geolocation do not contribute significantly to the results of our study shown at 200m or 600m resolutions."
- Figure 5: flight direction arrow added
- Figure 5, caption: added ",and the scene stretches approximately 37km along-track by 5km cross-track". (changed slightly from author response)
- Figure 6: caption: "150m" changed to "200m" (changed slightly from author response)
- Figure 9: Units added for Intensity (W m$^{-2}$ sr$^{-1}$ nm$^{-1}$) and Effective Radius (µm) axes
- Figure 9: added "of" and removed "reflectance"

The authors thank the three reviewers for their feedback and reports. Below is the markup of the manuscript with the reviewer comments taken into account. Markups are shown as red strikeouts with the correction inserted beforehand. Any added text that is not associated with a strikeout is in blue.

[revised manuscript text omitted]